# Nonnegative spatial factorization applied to spatial genomics

F. William Townes [1] ✉ & Barbara E. Engelhardt [2,3] ✉

Nonnegative matrix factorization (NMF) is widely used to analyze high-dimensional count data because, in contrast to real-valued alternatives such as factor analysis, it produces an interpretable parts-based representation. However, in applications such as spatial transcriptomics, NMF fails to incorporate known structure between observations. Here, we present nonnegative spatial factorization (NSF), a spatially-aware probabilistic dimension reduction model based on transformed Gaussian processes that naturally encourages sparsity and scales to tens of thousands of observations. NSF recovers ground truth factors more accurately than real-valued alternatives such as MEFISTO in simulations, and has lower out-of-sample prediction error than probabilistic NMF on three spatial transcriptomics datasets from mouse brain and liver. Since not all patterns of gene expression have spatial correlations, we also propose a hybrid extension of NSF that combines spatial and nonspatial components, enabling quantification of spatial importance for both observations and features. A TensorFlow implementation of NSF is available from https://github.com/willtownes/nsf-paper.

Spatially-resolved gene expression, or spatial transcriptomics (ST), has revolutionized the study of intact biological tissues[1]. ST data consist of discrete counts of transcript fragments from thousands of features (typically genes), along with the spatial coordinates of each observation (typically a cell). Since cell types are frequently unknown, dimension reduction is a vital tool for exploratory analysis.

Since ST data, just like single-cell RNA-sequencing (scRNA-seq), are high-dimensional read counts that map onto specific gene transcripts, in principle existing dimension reduction methods can be used to obtain a low-dimensional representation of gene expression at each spatial location. However, this application of scRNA-seq methods ignores the spatial coordinates that are the distinguishing feature of ST. We instead retain spatial locality information while performing dimension reduction on these data.

In contrast to standard approaches such as factor analysis (FA)[2], which ignore spatial context, MEFISTO[3] pioneered spatially-aware dimension reduction by representing high-dimensional gene expression features as a linear combination of a small number of latent Gaussian processes (GPs). A GP is a probability distribution over arbitrary functions on a continuous (for example, spatial) domain[4]. GPs are a fundamental tool in spatial statistics[5,6].

FA and MEFISTO use Gaussian likelihoods and produce real-valued factors. An alternative dimension reduction strategy, used by nonnegative matrix factorization (NMF), is to constrain the factors and their linear coefficients (loadings) to be nonnegative, which encourages sparsity and a more interpretable parts-based representation. For example, when applying dimension reduction to pixel-based representations of faces, NMF learns factors representing eyes, noses, mouths and ears. On the other hand, real-valued alternatives produce eigenfaces, or representations of whole faces, in each factor[7].

Here, we present nonnegative spatial factorization (NSF), a model for spatially-aware dimension reduction using an exponentiated GP prior over the spatial locations with a Poisson or negative binomial likelihood for count data. We also develop an NSF hybrid model (NSFH) that generalizes both NSF and probabilistic NMF to partition variability into both spatial and nonspatial sources. We illustrate the ability of

[1]Present address: Department of Statistics and Data Science, Carnegie Mellon University, Pittsburgh, PA, USA. [2]Present address: Data Science and Biotechnology Institute, Gladstone Institutes, San Francisco, CA, USA. [3]Present address: Department of Biomedical Data Science, Stanford University, Stanford, CA, USA. ✉e-mail: ftownes@andrew.cmu.edu; barbara.engelhardt@gladstone.ucsf.edu

nonnegative factorizations to identify a parts-based representation using simulations. We benchmark the different dimension reduction methods on three ST datasets, and interpret the biological relevance of spatial and nonspatial factors learned by NSFH on each. Our methods facilitate unsupervised identification of tissue regions, cell types and gene modules associated with biological processes, as well as interpolation of missing values. Additional background is provided in the Supplemental Introduction.

## Results

This paper proceeds as follows. We first define the generative models of FA, probabilistic NMF (PNMF), real-valued spatial factorization (RSF), NSF and NSFH. Second, we illustrate the ability of nonnegative factorizations to identify a parts-based representation using simulations. We then describe the basic features of the ST datasets and examine key results of a benchmarking comparison of different models. Next, we analyze three ST datasets from different technologies with NSFH and show how to interpret the spatial and nonspatial components. We conclude with a discussion of the implications of our results in ST data analysis and promising directions for future studies. Details on inference and parameter estimation, procedures for postprocessing nonnegative factor models and computing spatial importance scores along with data preprocessing are provided in the Methods and Supplemental Notes sections.

### Factor models for spatial count data

The data consist of a multivariate outcome $Y \in \mathbb{R}^{N \times J}$ and spatial coordinates $X \in \mathbb{R}^{N \times D}$. Let $i = 1, \ldots, N$ index the observations (for example, cells, spots or locations with a single $(x, y)$ coordinate value), $j = 1, \ldots, J$ index the outcome features (for example, genes), and $d = 1 \ldots, D$ index the spatial input dimensions.

**Nonspatial models.** In unsupervised dimension reduction such as principal components analysis (PCA), the goal is to represent $Y$ (or a normalized version $\tilde{Y}$ such as a mean-centered log of counts per million) as the product of two low-rank matrices $Y \approx FW'$, where the factors matrix $F$ has dimension $N \times L$ and the loadings matrix $W$ has dimension $J \times L$, with $L \ll J$. Let $l = 1, \ldots, L$ index the components. A probabilistic extension of PCA is FA:

$$\tilde{y}_{ij} \sim \mathcal{N}(\mu_{ij}, \sigma_j^2)$$

$$\mu_{ij} = \sum_{l=1}^{L} w_{jl} f_{il}$$

$$f_{il} \sim \mathcal{N}(m_l, s_l^2).$$

where the symbol ~ denotes 'distributed as'. A probabilistic version of NMF is PNMF:

$$y_{ij} \sim \text{Poi}(\nu_i \lambda_{ij})$$

$$\lambda_{ij} = \sum_{l=1}^{L} w_{jl} e^{f_{il}}$$

$$f_{il} \sim \mathcal{N}(m_l, s_l^2),$$

where $w_{jl} \geq 0$ and $\nu_i$ indicates a fixed size factor to account for differences in total counts per observation. In both of these unsupervised models, the prior on the factors $f_{il}$ assumes each observation is an independent draw and ignores spatial information $\mathbf{x}_i$.

**Spatial process factorization.** In spatial process factorization, we assume that spatially adjacent observations should have correlated outcomes. We encode this assumption via a GP prior over the factors.

We first consider real-valued spatial factorization (RSF), which we define as

$$\tilde{y}_{ij} \sim \mathcal{N}(\mu_{ij}, \sigma_j^2)$$

$$\mu_{ij} = \sum_{l=1}^{L} w_{jl} f_{il}$$

$$f_{il} = f_l(\mathbf{x}_i) \sim \text{GP}\left(\mu_l(\mathbf{x}_i), k_l(\mathbf{x}_i, X)\right),$$

where $\mu_l(\cdot)$ indicates a parametric mean function and $k_l(\cdot, \cdot)$ a positive semidefinite covariance (kernel) function. In our implementation, we specify the mean function as a linear function of the spatial coordinates,

$$\mu_l(\mathbf{x}_i) = \beta_{0l} + \mathbf{x}_i' \boldsymbol{\beta}_{1l}.$$

For the covariance function, we choose a Matérn kernel with fixed smoothness parameter 3/2. We allow each component $l$ to have its own amplitude and length-scale parameters that we estimate from data. RSF is a spatial analog to FA. MEFISTO has the same structure as RSF, but uses an exponentiated quadratic (EQ, also known as radial basis function or squared exponential) kernel instead of Matérn, and further places a sparsity-promoting prior on the loading weights $w_{jl}$. Our implementation is modular and can accept any positive semidefinite kernel. However, we found the Matérn kernel to have better numerical stability than the EQ in our experiments.

NSF is a spatial analog of PNMF.

$$y_{ij} \sim \text{Poi}(\nu_i \lambda_{ij})$$

$$\lambda_{ij} = \sum_{l=1}^{L} w_{jl} e^{f_{il}}$$

$$f_{il} = f_l(\mathbf{x}_i) \sim \text{GP}\left(\mu_l(\mathbf{x}_i), k_l(\mathbf{x}_i, X)\right).$$

For NSF, we use the same mean and kernel functions as RSF, but we additionally constrain the weights $w_{jl} \geq 0$.

We sought to quantify the relative importance of spatial versus nonspatial variation by combining NSF and PNMF into a semisupervised framework that we refer to as the NSF hybrid, or NSFH. NSFH consists of $L$ total factors, $T \leq L$ of which have spatial regularization and $L - T$ are nonspatial, and do not have spatial regularization. We recover NSF and PNMF as special cases when $T = L$ or $T = 0$, respectively. By default, we set $T = L/2$.

$$y_{ij} \sim \text{Poi}(\nu_i \lambda_{ij})$$

$$\lambda_{ij} = \sum_{l=1}^{T} w_{jl} e^{f_{il}} + \sum_{l=T+1}^{L} v_{jl} e^{h_{il}}$$

$$f_{il} = f_l(\mathbf{x}_i) \sim \text{GP}\left(\mu_l(\mathbf{x}_i), k_l(\mathbf{x}_i, X)\right)$$

$$h_{il} \sim \mathcal{N}(m_l, s_l^2).$$

Our implementations of PNMF, NSF and NSFH are modular with respect to the likelihood, so that the negative binomial or Gaussian distributions can be substituted for the Poisson. However, in our experiments we use the Poisson data likelihood.

**Postprocessing nonnegative factor models.** We postprocess fitted nonnegative models (PNMF, NSF and NSFH) by projecting factors and loadings onto a simplex. This highlights features (genes) that are enriched in particular components rather than those with high expression across all components. In the NSFH model, we interpret the ratio of loading weights for each feature across all spatial components as a spatial importance score. This score is analogous to the proportion of variance explained in PCA. In particular, a score of 1 means

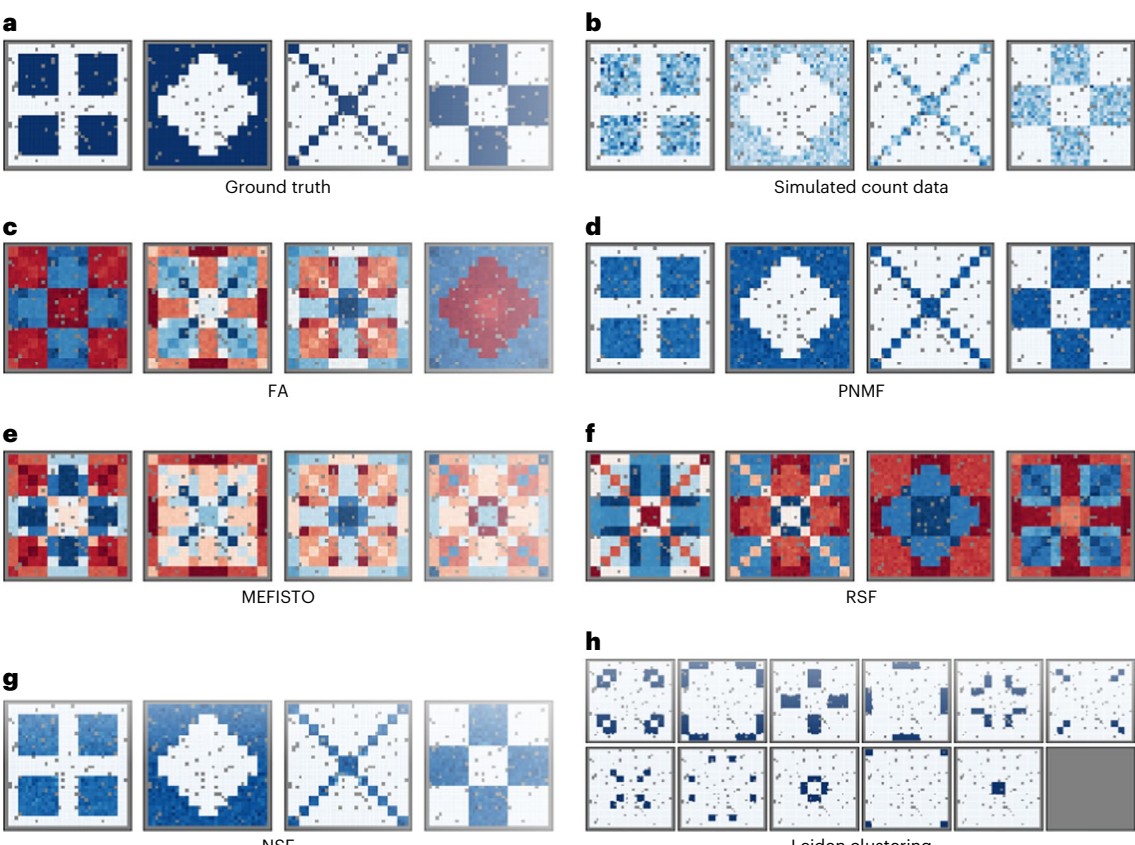

**Fig. 1 | Nonnegative factorizations recover a parts-based representation in 'quilt' simulated multivariate spatial count data. a**, Each of 200 features was randomly assigned to one of four nonnegative spatial factors. **b**, Negative binomial count data used for model fitting. **c**, Real-valued factors learned from unsupervised (nonspatial) dimension reduction. **d**, As **c** but using nonnegative components. **e**, Real-valued, spatially aware factors with EQ kernel. **f**, As **e** but with a Matérn kernel and without a sparsity-inducing prior. **g**, Nonnegative, spatially-aware factors. **h**, Unsupervised clustering of observations. Spatial models used all observations as IPs. Gray indicates observations held out for validation.

that variation in a gene's expression profile across all observations is completely captured by the spatial factors, whereas a 0 means that expression variation is completely captured by nonspatial factors. The gene-level scores can be used to identify spatially variable genes as pioneered by spatialDE[8]. We also compute observation-level scores by switching the role of the factors and loadings matrices; details are provided in the Methods.

**Simulations: nonnegativity and parts-based representation**
To illustrate the ability of nonnegative models to recover a parts-based factorization, we simulated multivariate count data from two sets of spatial patterns. The 'ggblocks' simulation was based on the Indian buffet process[9]. The true factors consisted of four simple shapes in different spatial regions. In the 'quilt' simulation, we created spatial patterns that overlapped in space. For both simulations, each of the 200 features was an independent negative binomial draw from one of the canonical patterns (Fig. 1a,b and Extended Data Fig. 1a,b). The simulated spatial patterns correspond to ground truth spatial factors. In fitting spatial models, we used all observations as inducing points (IPs) to maximize accuracy since the data were small enough to not pose an excessive computational burden.

Real-valued models FA and RSF estimated latent factors consisting of linear combinations of the true factors (Extended Data Fig. 1c,e,f and Fig. 1c,e,f). Nonnegative models PNMF and NSF identified each pattern as a separate factor (Extended Data Fig. 1d,g and Fig. 1d,g). Unsupervised clustering[10] is a special case of nonnegative factorization where the factors are constrained to be orthogonal[11]. This forces

each observation to belong to only one component. Like PNMF and NSF, Leiden clustering accurately identified spatially disjoint patterns in the ggblocks simulation (Extended Data Fig. 1h) but was unable to recognize overlapping spatial patterns in the quilt simulation (Fig. 1h). Overall, this demonstrates that the parts-based representation in PNMF is preserved in NSF.

To provide a quantitative assessment of model performance, we ran additional replicate simulations with different random seeds. In scenario I, we simulated counts based on quilt factors (four), ggblocks factors (four) or a concatenation of both (eight). We fitted each model with the true number of components. Nonnegative models PNMF and NSF learned factors and loadings with the greatest correlation to ground truth (Extended Data Fig. 2a,b, linear regression adjusting for simulation type: $t = 14.4$, $p < 2.2 \times 10^{-16}$ for factors and $t = 14.5$, $p < 2.2 \times 10^{-16}$ for loadings). Spatially aware models NSF and RSF had the lowest prediction error on held-out validation data (Extended Data Fig. 2c; $t = -14.9$, $p < 2.2 \times 10^{-16}$ controlling for simulation type).

In simulation scenario II, we assessed the ability of the hybrid model NSFH to distinguish spatial from nonspatial sources of variability. We used the same three generative models, but we concatenated additional nonspatial factors (three, three and six). We varied the loadings so that some of the features were purely spatial, some were purely nonspatial and some were mixed. NSFH exhibited lower prediction error compared to nonspatial PNMF and purely spatial NSF with the same number of components (Extended Data Fig. 3a; $t = -3.5$, $p \le 6.4 \times 10^{-4}$). We computed spatial importance scores for each feature based on the fitted NSFH model and compared to scores computed

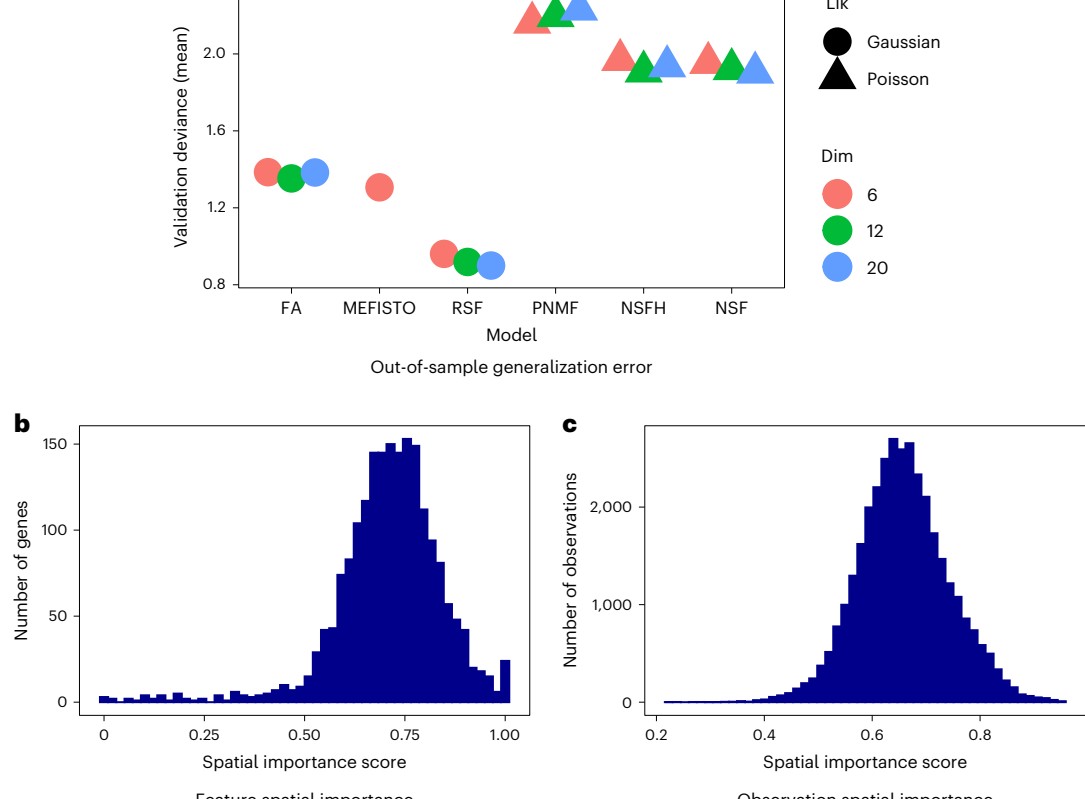

**Fig. 2 | Benchmarking spatial and nonspatial factor models on Slide-seqV2 mouse hippocampus spatial gene expression data. a**, Poisson deviance on held-out validation data. Lower deviance indicates better generalization accuracy. All spatial models used 2,000 IPs. MEFISTO could not be fit with more than six components due to out of memory errors. *Lik* represents likelihood, and *Dim* represents the number of latent dimensions (components). **b**, Each feature (gene) was assigned a spatial importance score derived from NSFH fit with 20 components (ten spatial and ten nonspatial). A score of 1 indicates spatial components explain all the variation. **c**, As **b** but with observations instead of features.

from the ground truth. NSFH was able to distinguish between spatial and nonspatial components with high accuracy across replicates (Extended Data Fig. 3b).

**Application to spatial transcriptomics (ST) datasets**

We examined the goodness of fit and interpretability of NSFs on three ST datasets (Supplementary Table 2). The Slide-seqV2 mouse hippocampus data[12] consist of 36,536 observations, each at a unique location. The XYZeq liver data[13] consist of 2,700 observations at 289 unique locations. Unlike the other protocols, each observation represents a single cell, but multiple cells are assigned to the same location. In other words, each spatial location in XYZeq contains multiple distinguishable observations, whereas in the other protocols each spatial location contains a single observation. Finally, the 10X Visium mouse brain data consist of 2,487 observations from an anterior sagittal section, each at a unique location.

Each protocol represents a different tradeoff between field of view (FOV) and spatial resolution. Slide-seqV2 has the smallest FOV and finest resolution, while XYZeq has the largest FOV and the coarsest resolution. Visium is intermediate in both criteria, capturing more spatial locations than XYZeq, but sacrificing the single-cell resolution of Slide-seqV2 with each observation representing an average of multiple nearby cells.

To assess the use of nonnegative and spatial factors in describing spatial sequencing data, we systematically compared all models (Supplementary Table 1) on all three datasets. We split each dataset randomly into a training set (95% of observations) and validation set (5% of observations), and we fitted each model with varying numbers

of components. We quantified goodness of fit using Poisson deviance between the observed counts in the validation data and the predicted mean values from each model fit to the training data; a small deviance indicates that the model fits the data well.

**Slide-seqV2 mouse hippocampus data.** On the Slide-seqV2 mouse hippocampus dataset, we benchmarked each model with $L = 6, 12, 20$ components and used $M = 2,000$ IPs. We additionally fit NSFH with $M = 3,000$ IPs for further downstream analyses. Real-valued factor models had lower validation deviance (higher generalization accuracy) than nonnegative models (Fig. 2a). Using linear regression to adjust for the number of components, we found a substantial difference between real-valued models and nonnegative models with respect to validation deviance ($t = -8.6, P \leq 1.9 \times 10^{-6}$). This was to be expected as real-valued factors can encode more information than nonnegative factors. The unsupervised models (FA and PNMF) had higher deviance than their spatially aware analogs (RSF, NSFH and NSF; $t = 10.9, P \leq 7.5 \times 10^{-7}$, adjusting for number of components and stratifying on real-valued versus nonnegative). RSF outperformed MEFISTO despite having nearly the same probabilistic structure. Using root mean squared error (RMSE) instead of Poisson deviance, we found the spatially aware models RSF, NSF and NSFH outperformed the nonspatial models FA and PNMF. The lowest RMSE was achieved by NSF (Extended Data Fig. 4a). We were unable to fit MEFISTO models with more than six components because they ran out of memory.

In terms of sparsity, MEFISTO had the highest fraction of zero entries in the loadings matrix due to its sparsity-promoting prior, followed by the nonnegative models NSFH, PNMF and NSF (Extended Data Fig. 5a).

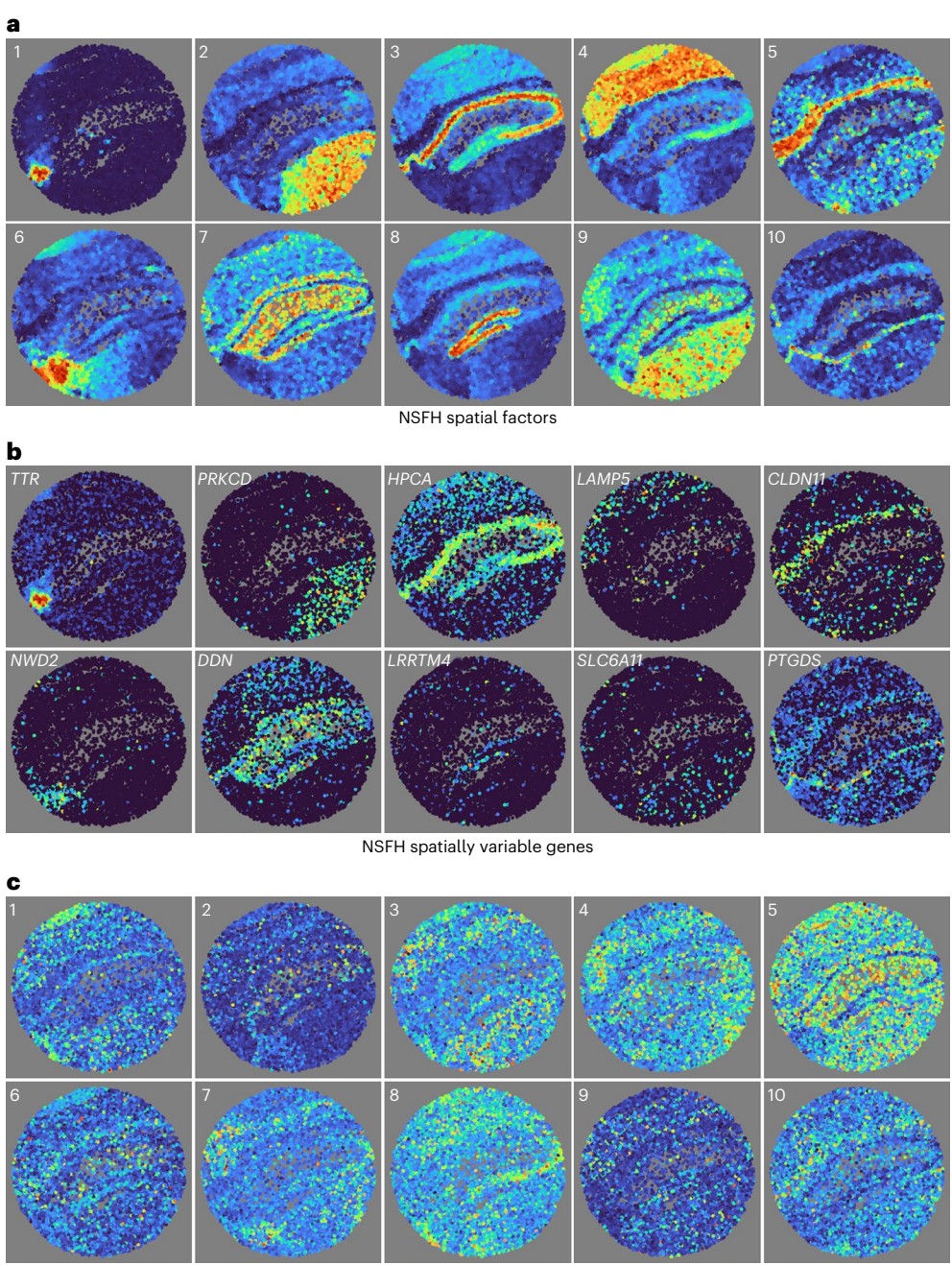

**Fig. 3 | NSFH combines spatial and nonspatial factors in Slide-seqV2 mouse hippocampus gene expression data.** FOV is a coronal section with left indicating the medial direction and right the lateral direction. **a**, Heatmap (red, high and blue, low) of square-root transformed posterior mean of ten spatial factors mapped into the $(x, y)$ coordinate space. **b**, As **a** but mapping expression levels of top genes with strongest enrichment to each spatial component. **c**, As **a** but mapping ten nonspatial factors from the same model.

This result was consistent across the other two datasets as well (Extended Data Fig. 6a,b). Increasing the number of components also increased the sparsity. The time to convergence was comparable for all spatial models, with nonspatial models converging substantially faster (Extended Data Fig. 5b). Among nonnegative models, the negative binomial likelihood took longer to converge but did not reduce generalization error (Extended Data Fig. 5c,d). Both NSF and NSFH had similar deviances, suggesting that including a mixture of spatial and nonspatial components (NSFH) did not degrade generalization in comparison to a strictly spatial model (NSF; one sided $t$-test $t = 0.53, p \leq 0.31$). NSFH also had better goodness of fit to training data than NSF (Extended Data Fig. 7a).

With variational GP models, one should generally maximize the number of IPs up to the number of training observations for optimal approximation to the posterior distribution. However, if the number of IPs is too large, the computational burden may become prohibitive. This was the case with the Slide-seqV2 data. We were able to fit MEFISTO with up to $M = 2,000$ IPs and our models with up to $M = 3,000$, relative to $N > 30,000$ observations. Increasing the number of IPs improved goodness of fit to training data in spatial models RSF and NSF with $L \in \{12, 20\}$ components (Extended Data Fig. 8a). However, for $L = 6$ or for models MEFISTO and NSFH, increasing IPs did not improve training fit. Further, the effect of IPs on prediction accuracy was ambiguous

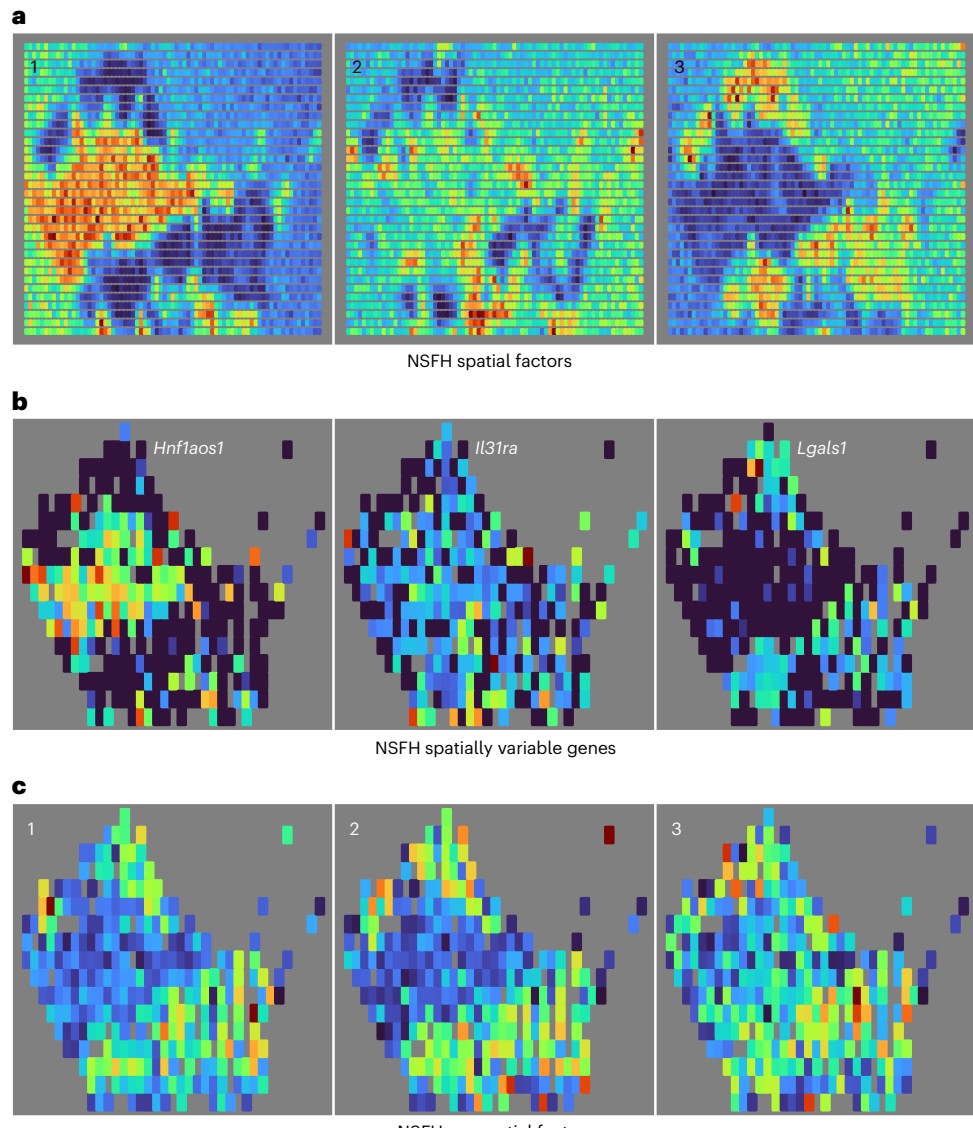

**Fig. 4 | NSFH model combines spatial and nonspatial factors in XYZeq mouse liver gene expression data. a**, Heatmap (red, high and blue, low) of square-root transformed posterior mean of three spatial factors mapped into the $(x, y)$ coordinate space. **b**, As **a** but mapping expression levels of top genes with strongest enrichment to each spatial component. **c**, As **a** but mapping three nonspatial factors from the same model.

across models (Extended Data Fig. 8b). As expected, increasing IPs consistently increased the computational burden of model fitting (Extended Data Fig. 8c).

We examined the biological relevance of nonnegative factorization by focusing on the NSFH model with $M = 3,000$ IPs and $L = 20$ components (ten spatial and ten nonspatial). Each factor was summarized by its (variational) approximate posterior mean. For each spatial factor, this is a function in the $(x, y)$ spatial coordinate system. For each nonspatial factor, the posterior is a vector with one value per observation. Spatial importance scores indicated that most genes were strongly spatially variable, although a small number were entirely nonspatial (Fig. 2b). At the observation level, spatial scores were less extreme, suggesting that both spatial and nonspatial factors are needed to explain cell state at each location (Fig. 2c). Spatial factors exhibited higher autocorrelation than nonspatial factors (Extended Data Fig. 9a).

Spatial factors mapped to specific brain regions (Fig. 3a) such as the choroid plexus (one), medial habenula (six) and dentate gyrus (eight). Even the thin meninges layer was distinguishable (ten), underscoring the high spatial resolution of the Slide-seqV2 protocol. Some of

these regions were also identified by other nonnegative models such as PNMF and by Leiden clustering, albeit less clearly (Supplementary Fig. 1a,b). Real-valued factor models FA and RSF were unable to identify distinct regions (Supplementary Fig. 2a,b).

We identified genes with the highest enrichment to individual components by examining the loadings matrix. Spatial gene expression patterns mirrored the spatial factors to which they were most associated (Fig. 3b). Nonspatial factors were generally dispersed across the FOV (Fig. 3c). Finally, we used the top genes for each component to identify cell types and biological processes (Supplementary Table 3) using scfind[14] and the Panglao database[15]. For example, spatial component 5 identified the corpus callosum, a white-matter region where myelination is crucial. Similarly, the top cell type for spatial component ten capturing the meninges layer was meningeal cells. Generally, neurons and glia were the most common cell types across all components. For comparison, we also clustered genes using Hotspot[16]. Of the 19 clusters, only six corresponded to neural biological processes (Supplementary Table 4). With a false discovery threshold of 0.05, Hotspot labeled all 2,000 variable genes as spatially variable. Using a spatial importance

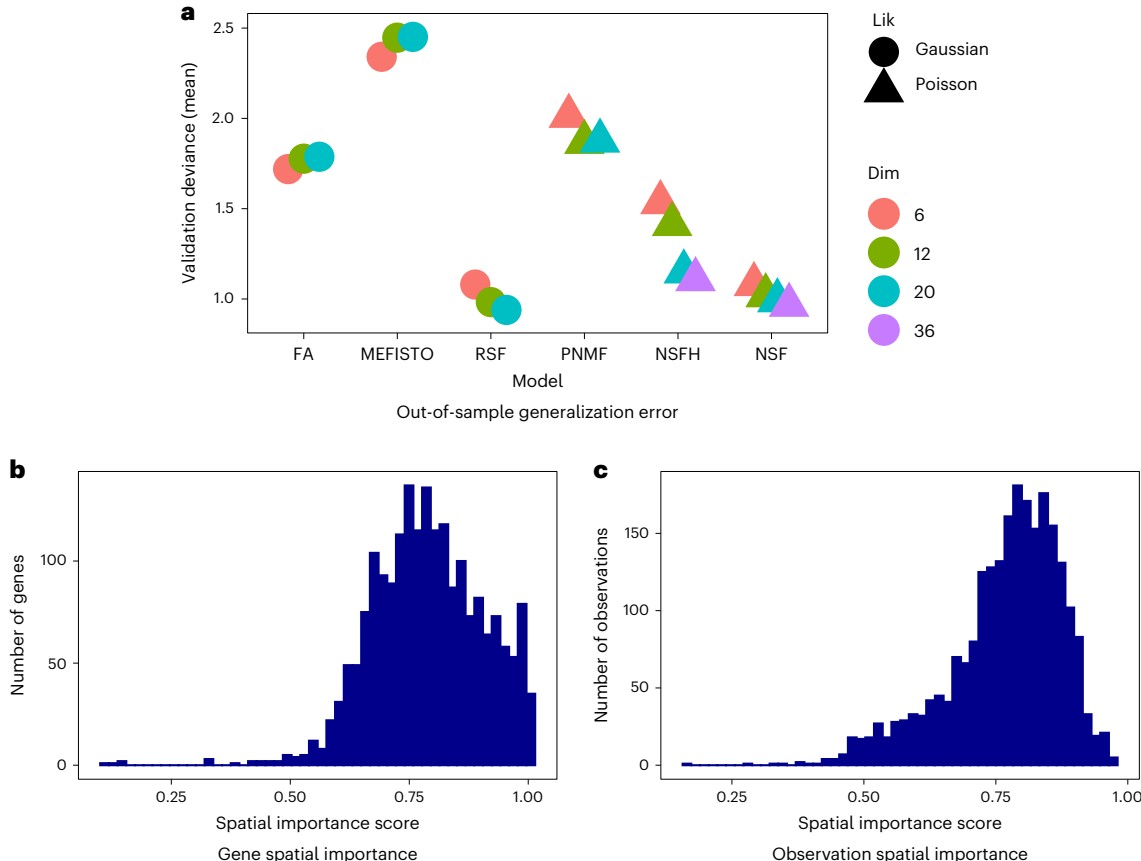

**Fig. 5 | Benchmarking spatial and nonspatial factor models on Visium mouse brain gene expression data. a**, Lower deviance indicates better generalization accuracy. **b**, Each feature (gene) was assigned a spatial importance score derived from NSFH fit with 20 components (ten spatial and ten nonspatial).

A score of 1 indicates spatial components explain all the variation. **c**, As **b** but with observations instead of features. All spatial models used 2,363 IPs. *Lik* represents likelihood, and *Dim* represents the number of latent dimensions (components).

score threshold of 0.5, NSFH identified 89 genes as nonspatial (Supplementary Fig. 3a). The Spearman rank correlation between scores from the two methods was 0.11.

**XYZeq mouse liver data.** On the XYZeq mouse liver dataset, we fit all models with $L$ = 6, 12, 20 components and used all 288 unique spatial locations as IPs. Real-valued factor models again had lower validation deviance than nonnegative models and spatial models again outperformed their nonspatial analogs (Extended Data Fig. 10a; linear regressions adjusting for number of components: $t$ = −25.3, $p \leq 1.9 \times 10^{-12}$ and $t$ = −9.6, $p \leq 2.8 \times 10^{-7}$, respectively.). The strictly spatial NSF model had slightly lower deviance than the hybrid spatial and nonspatial model NSFH (one sided $t$-test $t$ = 2.7, $p \leq 0.026$). The lowest predictive RMSE was again achieved by NSF, followed by NSFH and RSF (Extended Data Fig. 4b). NSFH again had better goodness of fit to training data than NSF (Extended Data Fig. 7b).

Focusing on the NSFH model with $M$ = 288 IPs and $L$ = 6 components, we found a strikingly bimodal distribution of spatial importance scores for both genes and observations (Extended Data Fig. 10b,c). Like the Slide-seqV2 data, most scores were greater than 0.5, suggesting spatial variation was more explanatory than nonspatial overall. Spatial factors again exhibited higher autocorrelation than nonspatial factors (Extended Data Fig. 9c). The first spatial factor identified normal liver tissue while the other spatial factors were associated with the tumor regions (Fig. 4a). Genes associated with spatial component 1 indicated an enrichment of hepatocytes, while genes in the other components were associated with immune cells (Fig. 4b and Supplementary Table 5). The nonspatial factors again showed no distinct spatial patterns

for these data (Fig. 4c), although they were associated with particular cell types and biological processes (Supplementary Table 5 and Supplementary Fig. 4). In contrast, only two of the six Hotspot gene clusters (likely macrophages) corresponded to relevant biological processes (Supplementary Table 6). Hotspot again labeled all 2,000 variable genes as spatially variable. In contrast, NSFH identified 412 genes as nonspatial (Supplementary Fig. 3b). The Spearman rank correlation between scores from the two methods was 0.29. Hepatocytes were enriched in the first component in PNMF, Leiden clustering, and FA (Supplementary Figs. 5a,b and 6a). Although this dataset contained many gaps between observations, NSFH spatial factors and RSF were able to learn continuous surfaces across the entire domain, in contrast to nonspatial alternatives (Fig. 4a and Supplementary Fig. 6b).

**Visium brain data.** On the Visium mouse brain data, we fit all models with $L$ = 6, 12, 20 components and used all 2,363 observations as IPs. We additionally fit NSF and NSFH with $L$ = 36 components. Goodness-of-fit results in terms of validation deviance were markedly different from the other two datasets (Fig. 5a). First, real-valued models did not dominate nonnegative models (linear regression adjusting for number of components, $t$ = 1.1, $p \leq 0.27$). Both NSF and NSFH had lower deviance than FA and MEFISTO. Linear regression on model categories with FA as baseline found MEFISTO higher ($t$ = 7.3, $p \leq 4.0 \times 10^{-6}$), and NSFH and NSF lower ($t$ = −5.5, $p \leq 8.0 \times 10^{-5}$ and $t$ = −8.9, $p \leq 3.8 \times 10^{-7}$, respectively). In fact, NSF had generalization accuracy comparable to the best-performing model (RSF, one sided $t$-test $t$ = 0.30, $p \leq 0.39$). Repeating the analysis with a larger validation set (20% versus 5%) did not alter these results (Supplementary Fig. 7). RSF had the lowest predictive RMSE, followed

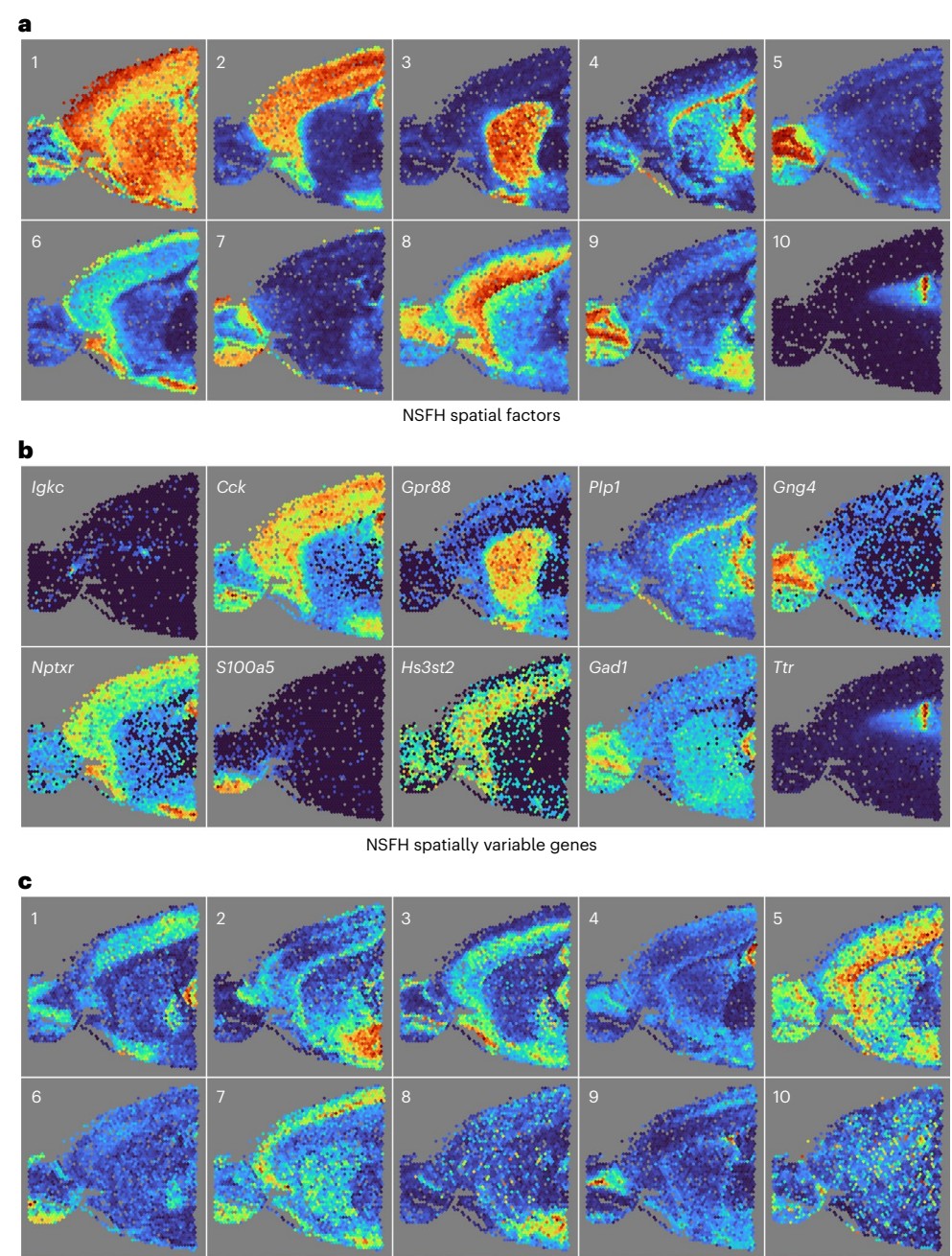

**Fig. 6 | NSFH model combines spatial and nonspatial factors in Visium mouse brain gene expression data.** FOV is a sagittal section with left indicating the anterior direction and right the posterior direction. **a**, Heatmap (red represents high, and blue represents low) of square-root transformed posterior mean of ten spatial factors mapped into the $(x, y)$ coordinate space. **b**, As **a** but mapping expression levels of top genes with strongest enrichment to each spatial component. **c**, As **a** but mapping ten nonspatial factors from the same model.

closely by NSF (Extended Data Fig. 4c). In terms of goodness of fit to training data, the results were different from the other two datasets. With the negative binomial likelihood, NSF outperformed NSFH. With the Poisson likelihood, the two models had similar deviances (Extended Data Fig. 7c).

It was necessary to increase the number of components in NSFH to reduce the deviance to a level comparable with NSF, whereas in the other datasets deviance did not vary substantially with the number of components and NSFH showed a similar performance to NSF. However, consistent with the other datasets, nonspatial models generally had higher deviance than their spatial analogs, reinforcing the importance

of including spatial information in out-of-sample prediction, interpolation and generalization (linear regression, adjusting for number of components, $t = 9.3$, $p \leq 1.5 \times 10^{-6}$).

Just as in the Slide-seqV2 dataset, there was a dramatic reduction in prediction error moving from MEFISTO to RSF (one sided $t$-test, $t = -25.7$, $p \leq 6.8 \times 10^{-6}$). This was surprising since these two models have very similar probabilistic structure. We hypothesized this difference could be due to the choice of spatial covariance function[17]: MEFISTO uses an EQ kernel whereas RSF, NSF and NSFH use a Matérn 3/2 kernel by default. To test this, we performed an extra analysis with RSF, NSF and NSFH combined with EQ. Using EQ led to more numerical instabilities

during optimization (Supplementary Table 7), but had no notable effect on predictive accuracy (Extended Data Fig. 8).

We next focused on interpretation of the NSFH model with $M = 2,363$ IPs and $L = 20$ components. Spatial importance scores indicate that spatial factors are most explanatory of variation at both the gene and observation level (Fig. 5b,c). Similar to the Slide-seqV2 data, most spatial factors mapped to specific brain regions (Fig. 6a) such as the cerebral cortex (two), corpus callosum (four) and choroid plexus (ten). The top genes for each spatial component again showed expression patterns overlapping with their associated factors (Fig. 6b). While most nonspatial factors were dispersed across the FOV (Fig. 6c), a few of them did exhibit spatial localization to areas such as the hypothalamus (two) and hippocampus (four). Some nonspatial factors also had high autocorrelation (Extended Data Fig. 9b), although increasing the number of components mitigated this phenomenon (Extended Data Fig. 9d). This illustrates that the nonspatial factors are not antagonistic to spatial variation but should be thought of as spatially naive or agnostic. Given that Visium does not provide single-cell resolution and this phenomenon was not observed in the other two datasets, we hypothesize that spatial patterns active in small numbers of observations may be more likely to be picked up as nonspatial factors under such conditions.

Using the top genes for each component, we identified cell types, brain regions and biological processes (Supplementary Table 8). For example, spatial component 3 aligned to the basal ganglia, and the top genes in this component are involved in the 'response to amphetamine' biological process. Nonspatial component 10 had many genes associated with erythroid progenitor cells. The nonspatial patterns in this component suggest that this factor includes cell types in blood; however, erythroid progenitor cells are not found in blood. We hypothesize these are actually erythrocytes, which have been shown to retain parts of the erythroid transcriptome despite the loss of the nucleus[18]. As in the Slide-seqV2 hippocampus data, neurons and glia were the most common cell types identified across all components. Similar patterns were detected by PNMF and Leiden clustering (Supplementary Fig. 9a,b), while FA and RSF factors were less distinct (Supplementary Fig. 10a,b). Hotspot identified 28 gene clusters, many of which were associated with relevant biological processes such as 'axonogenesis.' However, eight were associated with 'reproduction' and 'ribosomal large subunit assembly' (Supplementary Table 9). Hotspot again labeled all 2,000 variable genes as spatially variable. NSFH identified 19 genes as nonspatial (Supplementary Fig. 3c). The Spearman rank correlation between scores from the two methods was 0.26.

## Discussion

We present NSF, a probabilistic approach to spatially-aware dimension reduction on observations of count data based on Gaussian process regularization. We show how to combine spatial and nonspatial factors with the hybrid model NSFH. On simulated data, NSF, NSFH and the nonspatial model PNMF all recover an interpretable parts-based representation, whereas real-valued factorizations such as MEFISTO[3] capture a holistic embedding. A key advantage of spatially aware-factorizations over alternatives such as FA and PNMF is generalizability; spatial factor models learn latent functions over the entire spatial domain rather than only at the observed locations. On a benchmarking task using three ST datasets from three different technologies, NSF and NSFH had consistently lower out-of-sample prediction error than PNMF. Our implementation of RSF reduced prediction error compared to MEFISTO. We demonstrated how NSFH spatial and nonspatial components identify distinct regions in brain and liver tissue, cell types and biological processes. Finally, we quantified the proportion of variation explained by spatial versus nonspatial components at both the gene and observation level using spatial importance scores.

The choice of covariance function (kernel) is an important modeling choice in GPs[17]. We found that choosing a Matérn kernel instead of the EQ used by MEFISTO improved numerical stability. However, the difference

in kernels did not explain the improvement in accuracy of RSF compared to MEFISTO. Other features of RSF that differ from MEFISTO include the lack of sparsity-promoting prior in the loadings, inclusion of a linear mean function as part of the GP prior for each latent factor, optimization of hyperparameters by gradient descent instead of coordinate ascent and possibly stopping conditions or other implementation details.

All the spatial models we considered were based on linear combinations of GPs with variational inference using IPs[19,20]. Whereas this technique has improved GP scalability by enabling minibatching and nonconjugate likelihoods, the computational complexity still scales cubically with the number of IPs. A promising future direction for extending GPs to even larger numbers of observations is the nearest-neighbor approximation[21,22]. Further comments are provided in the Supplemental Discussion.

## Online content

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

## Methods

All spatial factor models were fit using variational inference with IPs[19,20]. Detailed mathematical derivations of objective functions, estimation and inference procedures are provided in the Supplemental Notes.

### Postprocessing nonnegative factorizations

Consider a generic nonnegative factorization $\Lambda = FW'$ or equivalently $\lambda_{ij} = \sum_l f_{il} w_{jl}$. We assume that the log-likelihood of data $Y$ depends on the $N \times L$ factors matrix $F$ and $J \times L$ loadings matrix $W$ only through $\Lambda$. The number of observations is $N$, number of components is $L$ and number of features is $J$. For notational simplicity, here we use $f_{il}$ to denote a nonnegative entry of a factor matrix rather than $e^{f_{il}}$ used in other sections. In the case that the model is probabilistic, we assume $f_{il}$ represents a posterior mean, posterior geometric mean or other point estimate. We project $F$, $W$ onto the simplex while leaving the likelihood invariant.

$$\bar{\mathbf{f}} = \sum_{i=1}^{N} F_{[i,:]} \in \mathbb{R}^L$$

$$F \leftarrow F \times \mathrm{diag}(\bar{\mathbf{f}})^{-1}$$

$$W \leftarrow W \times \mathrm{diag}(\bar{\mathbf{f}})$$

$$\bar{\mathbf{w}} = \sum_{l=1}^{L} W_{[:,l]} \in \mathbb{R}^J$$

$$W \leftarrow \mathrm{diag}(\bar{\mathbf{w}})^{-1} \times W.$$

Note that after this transformation $\Lambda = FW' \times \mathrm{diag}(\bar{\mathbf{w}})$. We now have that the columns of $F$ all sum to one and the rows of $W$ all sum to one (that is, they lie on the simplex). In the ST context, the features are genes. A particular row of $W$ represents a single gene's soft clustering assignment to each of the $L$ components. If $w_{jl} = 1$ this meant all of that gene's expression could be predicted using only component $l$, whereas if $w_{jl} = 0$ this meant that component $l$ was irrelevant to gene $j$. For a given component $l$, we identified the top associated genes by sorting the $w_{jl}$ values in decreasing order.

We refer to the above procedure as 'SPDE-style' postprocessing due to its similarity to spatialDE[8]. An alternative postprocessing scheme is 'LDA-style'[23,24] where the roles of $F$ and $W$ are switched. This results in a loadings matrix whose columns sum to one ('topics') and a factors matrix whose rows sum to one. LDA-style postprocessing provides a soft clustering of observations instead of features. We used SPDE-style postprocessing throughout this work with the sole exception of computing spatial importance scores for observations, described below.

**NSFH spatial importance scores.** Let $F \in \mathbb{R}_+^{N \times T}$ represent the spatial factors matrix (rather than $e^F$), with corresponding loadings $W \in \mathbb{R}_+^{J \times T}$. Similarly, let $H \in \mathbb{R}_+^{N \times (L-T)}$ represent the nonspatial factors (rather than $e^H$) with corresponding loadings $V \in \mathbb{R}_+^{J \times (L-T)}$. Let $A = [F, H] \in \mathbb{R}_+^{N \times L}$ and $B = [W, V] \in \mathbb{R}_+^{J \times L}$.

To obtain spatial importance scores for features (genes), we applied SPDE-style postprocessing to $A$, $B$. The score $\gamma_j$ for feature $j$ is given by the sum of the loading weights across all the spatial components.

$$W \leftarrow B_{[:,1:T]}$$

$$\gamma_j = \sum_{l=1}^{T} w_{jl}.$$

Due to the initial postprocessing, $0 \le \gamma_j \le 1$ for all $j$. If $\gamma_j = 0$ then all the variation in feature $j$ was explained by the nonspatial factors. If $\gamma_j = 1$ then all the variation was explained by the spatial factors.

To obtain spatial scores for observations, we applied LDA-style postprocessing to $A$, $B$. The score $\rho_i$ for observation $i$ is given by the sum of the factor values across all the spatial components.

$$F \leftarrow A_{[:,1:T]}$$

$$\rho_i = \sum_{l=1}^{T} f_{il}.$$

As before, $0 \le \rho_i \le 1$ for all $i$. If $\rho_i = 0$ then all the variation in observation $i$ was explained by the nonspatial factors. If $\rho_i = 1$ then all the variation was explained by the spatial factors.

### Initialization

Real-valued models were initialized with singular value decomposition. Nonnegative models were initialized with the scikit-learn implementation of NMF[25]. For NSFH, we sorted the initial NMF factors and loadings in decreasing order of spatial autocorrelation using Moran's I statistic[26] as implemented in Squidpy[27]. The first $T$ factors were assigned to the spatial component and the remaining $L - T$ factors to the nonspatial component.

### Detecting convergence of stochastic optimizers

Since the objective functions of the models considered here could only be evaluated through sampling, the trace of each optimization exhibited random fluctuations. This precluded detection of convergence by simple relative or absolute thresholding of changes in the current versus the previous iteration. Rather than simply run each optimization a fixed number of iterations, we adopted a smoothing strategy. Starting at iteration 110 and repeating every ten iterations, we fit a cubic polynomial to the most recent 100 values of the objective function. Since this was a linear smoother, we could then evaluate the smoothed objective function values with negligible computational overhead. We declared convergence and stopped optimization whenever the relative change between smoothed current and smoothed previous objective function value was less than $5 \times 10^{-5}$.

### Simulations

**Scenario I: spatial components only.** To provide illustrative examples, we simulated multivariate counts with spatial correlation patterns. In the ggblocks and quilt simulations, each latent factor representing a canonical spatial pattern consisted of a $36 \times 36$ grid of locations ($N = 1,296$ total spatial locations). The number of features ('genes') was set to $J = 200$. Each feature was randomly assigned to one of the four patterns with uniform probabilities. Entries of the $1,296 \times 200$ mean matrix were set to 20.2 in the active region (where a shape is visible) and 0.2 elsewhere. To accomplish this, we defined a $200 \times 4$ binary loadings matrix $W$ where entry $w_{jl} = 20$ if feature $j$ is assigned to pattern $l$ and zero otherwise. We represented each spatial pattern as a column in a $1,296 \times 4$ factors matrix $F$ where $f_{il} = 1$ if spatial location $i$ is active in pattern $l$ and $f_{il} = 0$ otherwise. The counts $Y$ were then drawn from a negative binomial distribution with mean $M = 0.2 + FW'$ and shape parameter 10 to promote overdispersion. A random subset of 65 observations (5%) was withheld for validation leaving 1,231 for training. For MEFISTO, RSF and FA the count data were normalized to have the same total count at each spatial location, then log transformed with a pseudocount of one. Features were centered before applying each dimension reduction method. For PNMF and NSF the raw counts were used as input. The unsupervised methods (PNMF and FA) used only the $1,231 \times 200$ count matrix, while the supervised methods (NSF, RSF and MEFISTO) also used the $1,231 \times 2$ matrix of spatial coordinates. Since this was a smaller dataset, all spatial coordinates were used as IP locations to maximize accuracy. All models were fit with $L = 4$ components.

For quantitative benchmarking, we created five replicates each of three generative models: quilt, ggblocks and both. Quilt and ggblocks followed the same procedure described above but varying random

seeds. The 'both' model concatenated the ggblocks and quilt factors to have a larger number of total factors (eight instead of four). We fit five models to each of the 15 replicates: FA, MEFISTO, RSF, PNMF and NSF.

To assess model performance, we computed several metrics based on the fitted models. First, we computed the predictive accuracy on the held-out validation data by taking the average across observations of the Poisson deviance between the predicted mean from each model and the true count value. A low deviance indicated better generalization accuracy. We also examined the correlation between fitted factors and true factors from the generative model. For each true factor, we computed Pearson correlation to all fitted factors and identified the factor with the highest absolute correlation. For each model and replicate, we then summarized these correlations by taking the minimum across factors. If the minimum correlation was high, this indicated that all the true factors were captured accurately by the fitted model. The same process was repeated to assess accuracy of fitted loadings matrices.

**Scenario II: spatial and nonspatial components.** In simulation scenario II, we assessed the ability of the hybrid model NSFH to separate spatial and nonspatial factors compared to purely spatial NSF and purely nonspatial PNMF. We generated $T = 4, 4, 8$ spatial factors $F$ following the ggblocks, quilt and both patterns, respectively. As before this led to $N = 36^2 = 1,296$ total observations. The spatial factors were assigned values of $f_{il} = 1$ in active regions and zero elsewhere. We then generated three nonspatial factors $H$ in the ggblocks and quilt scenarios and six in the both scenario. Each element $h_{il}$ of the nonspatial factor matrices was drawn independently from a Bernoulli distribution with probability 0.2.

We simulated $J = 500$ features according to either a split pattern or a mixed pattern. In the split pattern, half of the features (250) were assigned to spatial components and the remaining 250 to the nonspatial components. Using the ggblocks simulation as an example, the first 250 rows of the $500 \times 4$ spatial loadings matrix $W$ would therefore have exactly one entry with a value of 20 and the rest as zeros. The distribution of the active (nonzero) entries was drawn from a uniform distribution. The nonspatial loadings matrix $V$ would be all zeros in the first 250 rows and the last 250 rows would have exactly one entry per row with value 20, again drawn uniformly.

In the mixed pattern, all 500 genes were assigned to both a spatial component and a single nonspatial component. Again, using ggblocks as an example, every row of the spatial loadings matrix $W$ would have exactly one entry with a value of 12 and zeros elsewhere. Each row of the nonspatial loadings matrix $V$ would have exactly one entry with value 8 and zeros elsewhere. As before the active entries were drawn uniformly at random for each row.

Finally, in both scenarios the $1,296 \times 500$ mean matrix was defined as $M = 0.2 + FW' + HV'$ (that is, the background mean in inactive regions was set to 0.2). The counts were drawn from negative binomial distribution with shape parameter 10 as before. We repeated this process with five different random seeds leading to 15 replicates of the split scenario and 15 replicates of the mixed scenario.

We then fit PNMF, NSF and NSFH to each replicate with the correct number of factors, holding out a random 5% of observations for validation. As before we assessed predictive accuracy for each model with average Poisson deviance. We computed spatial importance scores for each feature (a vector of length $J = 500$) according to the ground truth factors and loadings and compared the scores to the fitted models using Euclidean distance. Note that for the split scenario, the true scores were close to either zero or one, while for the mixed scenario, the true scores were close to 0.6. For PNMF, spatial importance scores were always zero, while for NSF scores were always one for all features, representing two limiting special cases of NSFH where $T = 0$ and $T = L$, respectively.

## Data acquisition and preprocessing
For all datasets, after quality control filtering of observations, we selected the top 2,000 informative genes using Poisson deviance as

a criterion[28,29]. Raw counts were used as input to nonnegative models (NSF, PNMF, NSFH) with size factors computed by the default Scanpy method as described below[30]. For real-valued models with Gaussian likelihoods (RSF, FA, MEFISTO), we followed the default Scanpy normalization for consistency with MEFISTO. The raw counts were normalized such that the total count per observation equaled the median of the total counts in the original data. The normalized counts were then log transformed with a pseudocount of one, and the features were centered to have mean zero. This scaled, log-normalized version of the data was then used for model fitting.

**Visium mouse brain.** The dataset 'Mouse Brain Serial Section 1 (Sagittal-Anterior)' was downloaded from https://support.10xgenomics.com/spatial-gene-expression/datasets. To facilitate comparisons, preprocessing followed the MEFISTO tutorial (https://nbviewer.jupyter.org/github/bioFAM/MEFISTO_tutorials/blob/master/MEFISTO_ST.ipynb)[3]. Observations (spots) with total counts fewer than 100 or mitochondrial counts greater than 20% were excluded.

**Slide-seqV2 mouse hippocampus.** This dataset was originally produced by ref. 12. We obtained it through the SeuratData R package[31] and converted it to a Scanpy H5AD file[30] using SeuratDisk[32]. Observations (spots) with total counts fewer than 100 or mitochondrial counts greater than 20% were excluded.

**XYZeq mouse liver.** This dataset was originally produced by ref. 13. We obtained it from the Gene Expression Omnibus[33], accession number GSE164430. We focused on sample liver_slice_L20C1, which was featured in the original publication, and downloaded it as an H5AD file. The spatial coordinates were provided by the original authors. We did not exclude any observations (cells), since all had total counts greater than 100 and mitochondrial counts fewer than 20%.

**Significance testing.** Linear models (regressions and $t$-tests) were used to quantify the statistical significance of comparisons in main results. $P$ values were reported without adjustment for multiple testing.

## Clustering
Clustering of observations followed the Scanpy[30] tutorial https://scanpy-tutorials.readthedocs.io/en/latest/spatial/basic-analysis.html, which uses the Leiden algorithm[10]. Clustering of features followed the Hotspot[16] tutorial https://hotspot.readthedocs.io/en/latest/Spatial_Tutorial.html using $K = 20$ nearest neighbors.

## Cell types and Gene Ontology terms
For each dataset, we fit a NSFH model and applied SPDE-style post-processing such that the loadings matrices had rows (representing genes) summing to one across all components. We then examined each column of the loadings matrix (representing a component) and identified the five genes with largest weights. We then manually searched for cell types on scfind (https://scfind.sanger.ac.uk/)[14]. If no results were found, we next searched the Panglao database (https://panglaodb.se)[15]. We identified brain regions in the Slide-seqV2 hippocampus and Visium brain datasets by referring to the interactive Allen Brain Atlas (https://atlas.brain-map.org)[34]. Gene Ontology annotations for all genes were downloaded from the BioMart ENSEMBL database (release 104, May 2021) using the biomaRt package (v.2.48.0) in Bioconductor (v.3.13). Enriched terms were identified using the topGO Bioconductor package (v.2.44.0) with the default algorithm 'weight01' and statistic 'fisher', considering the top 100 genes (with largest weights in the loadings matrix) for each component against a background of all other genes. The same parameters were used in searching for biological processes associated with Hotspot clusters.

## Software versions

We implemented all models using Python v.3.8.10, tensorflow v.2.5.0, tensorflow probability v.0.13.0. Other Python packages used include scanpy v.1.8.0, Squidpy v.1.1.0, scikit-learn v.0.24.2, pandas v.1.2.5, numpy v.1.19.5 and scipy v.1.7.0. We used the MEFISTO implementation from the mofapy2 Python package, installed from the GitHub development branch at commit 8f6ffcb5b18d22b3f44ff2a06bcb92f2806afed0. Graphics were generated using either matplotlib v.3.4.2 in Python or ggplot2 v.3.3.5 (ref. 35) in R (v.4.1.0). The R packages Seurat v.0.4.3 (ref. 36), SeuratData v.0.2.1 and SeuratDisk v.0.0.0.9019 were used for some initial data manipulations. Computationally intensive model fitting was done on Princeton's Della cluster. Each model was assigned 12 CPU cores. We provided the following total memory per dataset: 180 Gb for Slide-seq V2, 72 Gb for Visium and 48 Gb for XYZeq.

## Reporting summary

Further information on research design is available in the Nature Portfolio Reporting Summary linked to this article.

## Data availability

All data used in this paper are from public sources. The Visium mouse brain dataset Mouse Brain Serial Section 1 (Sagittal-Anterior) was downloaded from https://cf.10xgenomics.com/samples/spatial-exp/1.1.0/V1_Mouse_Brain_Sagittal_Anterior/V1_Mouse_Brain_Sagittal_Anterior_filtered_feature_bc_matrix.h5 and https://cf.10xgenomics.com/samples/spatial-exp/1.1.0/V1_Mouse_Brain_Sagittal_Anterior/V1_Mouse_Brain_Sagittal_Anterior_spatial.tar.gz. The Slide-seqV2 mouse hippocampus dataset was originally produced by ref. 12. We obtained it through the SeuratData R package (v.0.2.1)[31]. Raw data are available from https://singlecell.broadinstitute.org/single_cell/study/SCP815/highly-sensitive-spatial-transcriptomics-at-near-cellular-resolution-with-slide-seqv2#study-summary. The XYZeq mouse liver data were obtained from the Gene Expression Omnibus[33], accession number GSE164430. We focused on sample liver_slice_L20C1, which was featured in the original publication, and downloaded it as an H5AD file. The spatial coordinates were provided by the original authors. Additional databases used in the study included Biomart (http://www.biomart.org/), Panglao (https://panglaodb.se/) and scfind (https://scfind.sanger.ac.uk/).

## Code availability

Code for reproducing the analyses of this paper is available under LGPL-3.0 license from https://github.com/willtownes/nsf-paper or from Zenodo DOI https://doi.org/10.5281/zenodo.7130878 (ref. 37). An installable Python package is available from https://github.com/willtownes/spatial-factorization-py. Further information on research design is available in the Nature Research Reporting Summary linked to this article.

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

## Acknowledgements

We thank B. Velten and the MEFISTO team for assistance in model fitting and responsiveness in bug fixes. G. Hartoularos and D. Cable gave helpful advice on data acquisition and processing. A. Minerva contributed expertise in neuroanatomy. T. Townes helped with graphics. A. Wu, S. Keeley, S. Hicks and K. Hansen, along with members of the 'BEEHIVE' including A. Verma, D. Li, A. Jones, D. Cai and S.D. Ang provided valuable insight through informal discussions. F.W.T. and B.E.E. were funded by Helmsley Trust grant nos. AWD1006624, NIH NCI 5U2CCA233195, NIH NHLBI R01 HL133218 and NSF CAREER AWD1005627.

The work reported on in this paper relied on Princeton Research Computing resources at Princeton University, which is a consortium led by the Princeton Institute for Computational Science and Engineering and the Office of Information Technology's Research Computing.

## Author contributions

B.E.E. and F.W.T. conceived the project. F.W.T. implemented the model. B.E.E. and F.W.T. analyzed the data. F.W.T. generated the figures. B.E.E. and F.W.T. wrote the paper.

## Competing interests

B.E.E. is on the SAB for Creyon Bio, ArrePath and Freenome. B. E. E. consults for Neumora and Cellarity. The remaining author declares no competing interests.

## Additional information

**Extended data** is available for this paper at https://doi.org/10.1038/s41592-022-01687-w.

**Correspondence and requests for materials** should be addressed to F. William Townes or Barbara E. Engelhardt.

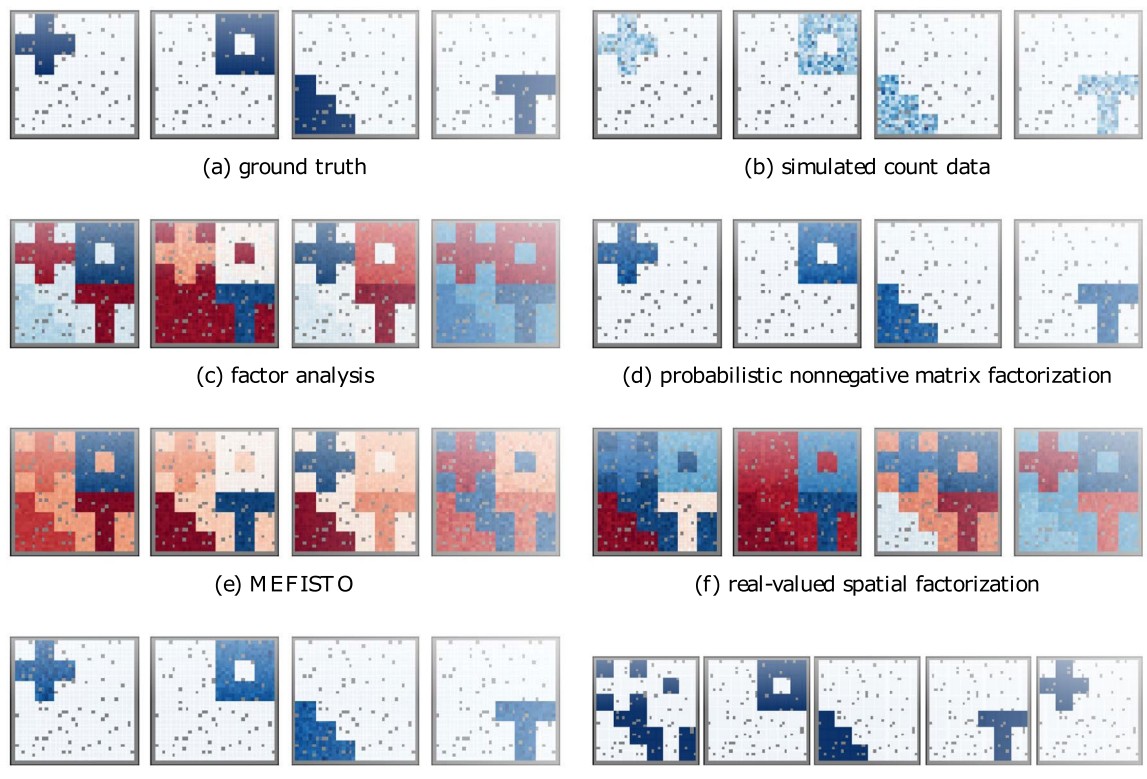

(a) ground truth

(b) simulated count data

(c) factor analysis

(d) probabilistic nonnegative matrix factorization

(e) MEFISTO

(f) real-valued spatial factorization

(g) nonnegative spatial factorization

(h) Leiden clustering

**Extended Data Fig. 1 | Nonnegative factorizations recover a parts-based representation in "ggblocks" simulated multivariate spatial count data.** (a) Each of 200 features was randomly assigned to one of four nonnegative spatial factors. (b) Negative binomial count data used for model fitting. (c) Real-valued factors learned from unsu- pervised (nonspatial) dimension reduction. (d) as (c) but using nonnegative components. (e) Real-valued, spatially aware factors with exponentiated quadratic (EQ) kernel. (f) as (e) but with Matern kernel and without sparsity-inducing prior. (g) Nonnegative, spatially-aware factors. (h) Unsupervised clustering of observations. Spatial models used all observations as inducing points. Gray indicates observations held out for validation.

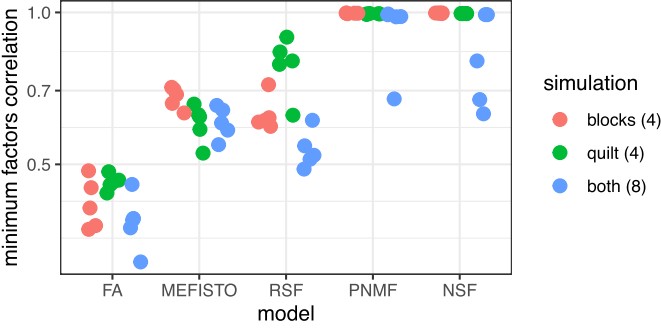

(a) factors correlation

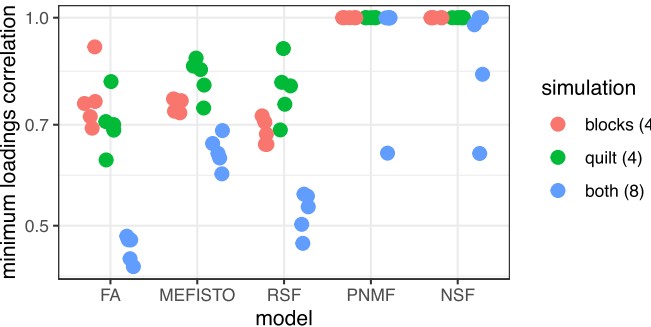

(b) loadings correlation

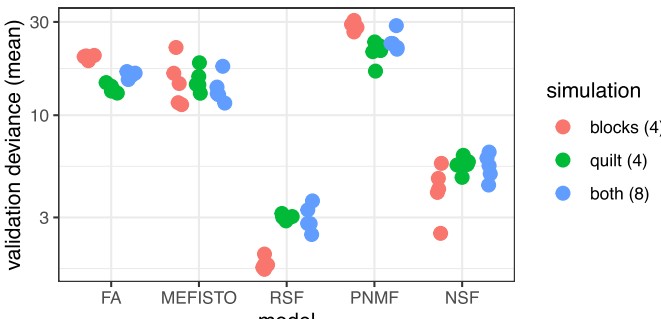

(c) prediction error

**Extended Data Fig. 2 | Benchmarking spatial and nonspatial factor models on simulation scenario I.** (a) Nonnegative models PNMF and NSF closely matched ground truth. Each true factor was aligned by Pearson correlation to the closest matching factor in each fitted model and the minimum correlation across all factors was computed for each model and simulation replicate. Higher minimum correlations indicate more accurate models. (b) as (a) but using correlations between loadings matrices. (c) Spatially-aware models NSF and RSF had best prediction accuracy (lowest Poisson deviance) on held-out validation data. FA: factor analysis, RSF: real-valued spatial factorization, PNMF: probabilistic nonnegative matrix factorization, NSF: nonnegative spatial factorization. Spatial models used all observations as inducing points.

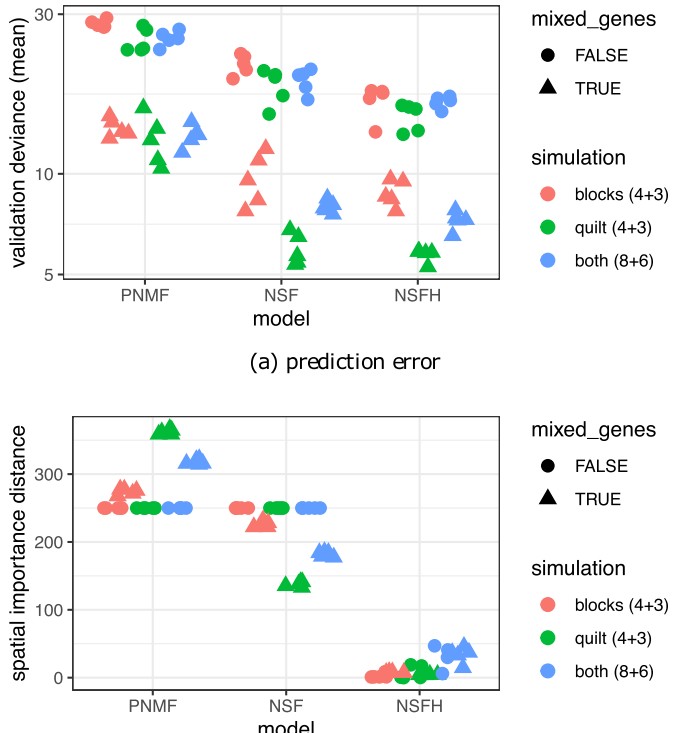

**Extended Data Fig. 3 | Benchmarking spatial, nonspatial, and hybrid factor models on simulation scenario II.** (a) Nonnegative spatial factorization hybrid (NSFH) model has highest generalization accuracy (lowest Poisson deviance prediction error) compared to purely nonspatial probabilistic nonnegative matrix factorization (PNMF) and or nonnegative spatial factorization (NSF). (b) NSFH spatial importance scores per feature are closest to scores computed from ground truth loadings. Spatial models used all observations as inducing points. Mixed genes is true for simulations where features have loadings on both spatial and nonspatial components. When mixed genes is false, a feature is assigned to be either strictly spatial or strictly nonspatial.

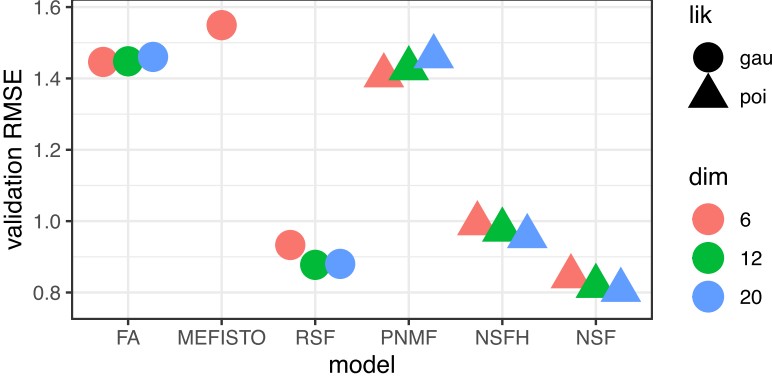

(a) Slide-seqV2 mouse hippocampus

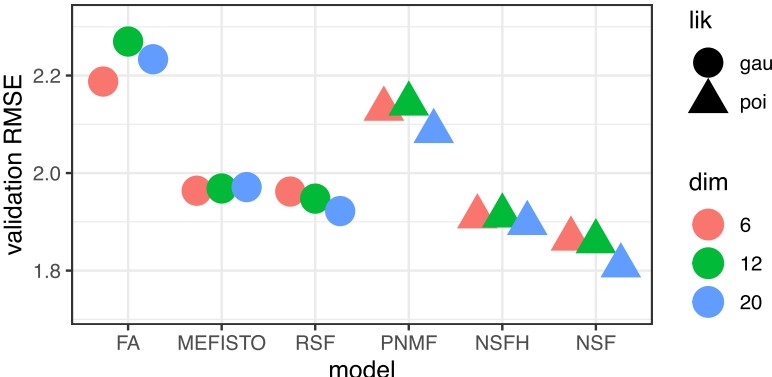

(b) XYZeq mouse liver/tumor

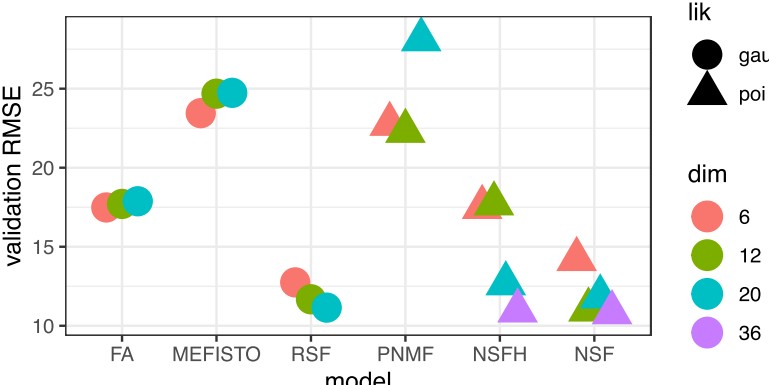

(c) Visium mouse brain

**Extended Data Fig. 4 | Comparison of predictive performance of spatial and nonspatial factor models on real datasets.** RMSE: root mean squared error on held-out observations, dim: number of latent dimensions or components, FA: factor analysis, RSF: real-valued spatial factorization, PNMF: probabilistic nonnegative matrix factorization, NSF: nonnegative spatial factorization, NSFH: NSF hybrid model, lik: likelihood, gau: Gaussian, poi: Poisson.

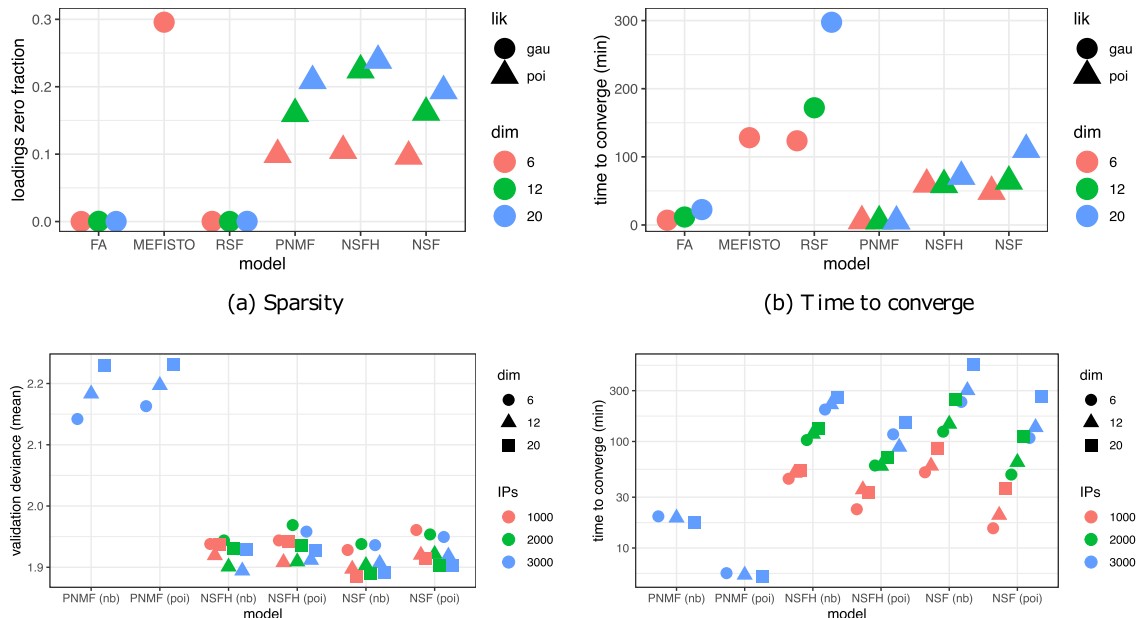

(a) Sparsity

(b) Time to converge

(c) Validation deviance by likelihood

(d) Time to converge by likelihood

**Extended Data Fig. 5 | Benchmarking spatial and nonspatial factor models on Slide-seqV2 mouse hippocampus gene expression data.** FA: factor analysis, RSF: real-valued spatial factorization, PNMF: probabilistic nonnegative matrix factorization, NSF: nonnegative spatial factorization, NSFH: NSF hybrid model, lik: likelihood, gau: Gaussian, poi: Poisson, nb: negative binomial. (a) Sparsity of loadings matrix increases with larger numbers of components (dim) in nonnegative models PNMF, NSFH, and NSF. (b) Nonnegative spatial models NSF and NSFH converge faster than MEFISTO but not as fast as nonspatial models FA and PNMF. (c) Negative binomial and Poisson likelihoods provide similar generalization accuracy (lower deviance) in nonnegative models. (d) Negative binomial likelihood is more computationally expensive than Poisson likelihood in nonnegative models.

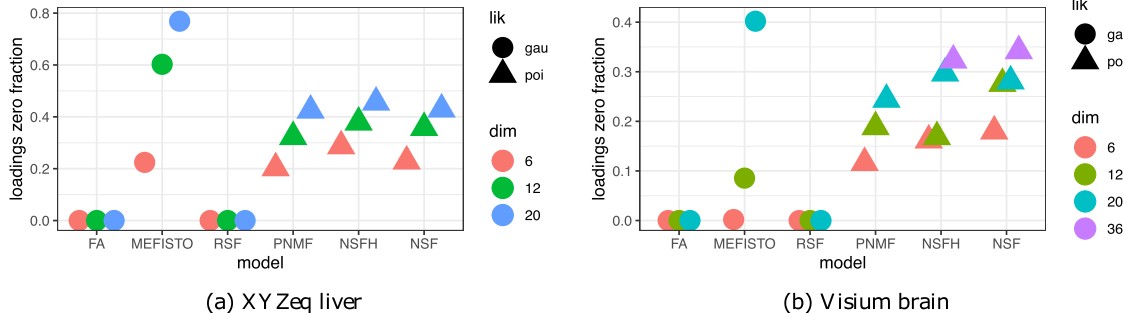

(a) XYZeq liver

(b) Visium brain

**Extended Data Fig. 6 | Sparsity of loadings matrices.** Sparsity increases with larger numbers of components (dim) in nonnegative models PNMF, NSFH, and NSF as well as real-valued model MEFISTO. (a) XYZeq mouse liver/tumor dataset. (b) Visium brain dataset.

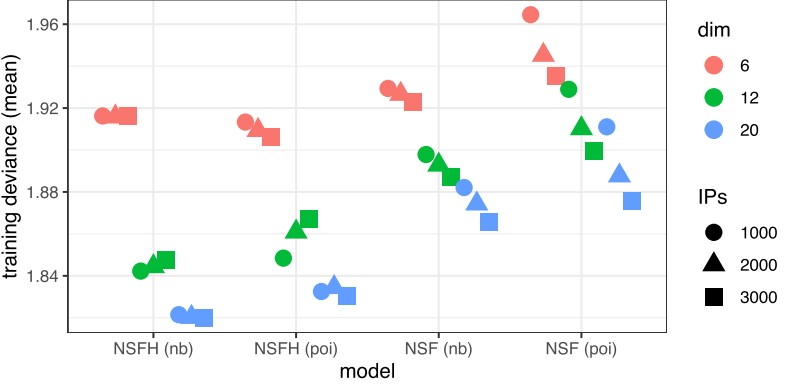

(a) Slide-seqV2 mouse hippocampus

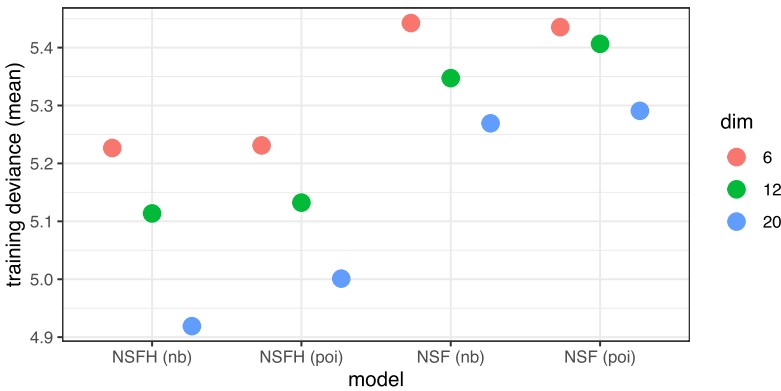

(b) XYZeq mouse liver/tumor

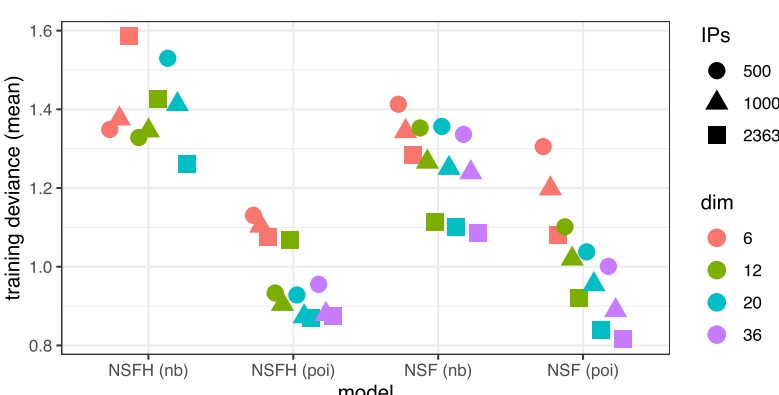

(c) Visium mouse brain

**Extended Data Fig. 7 | Goodness-of-fit of nonnegative spatial factorization (NSF) and NSF hybrid model (NSFH) to real datasets.** Lower deviance indicates better fit to training data. dim: number of latent dimensions or components, IPs: number of inducing points, lik: likelihood, poi: Poisson, nb: negative binomial. For XYZeq, all 288 unique spatial locations were used as IPs.

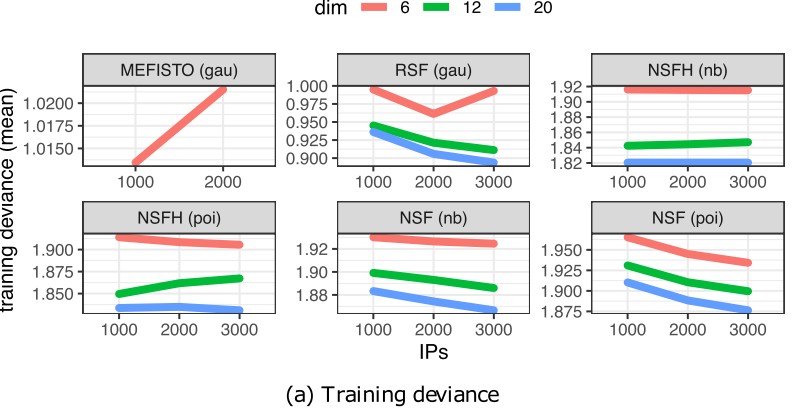

(a) Training deviance

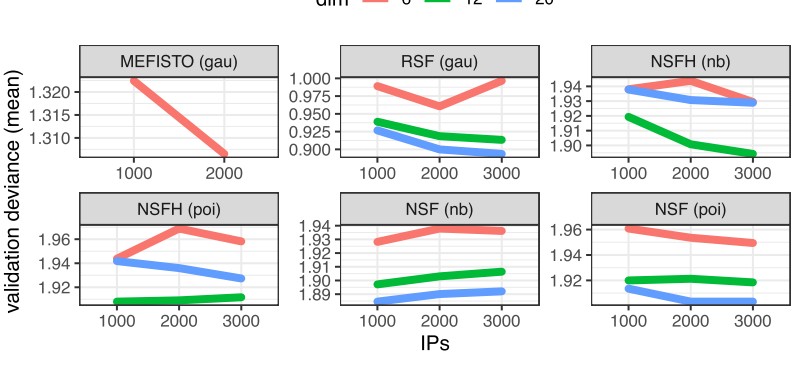

(b) Validation deviance

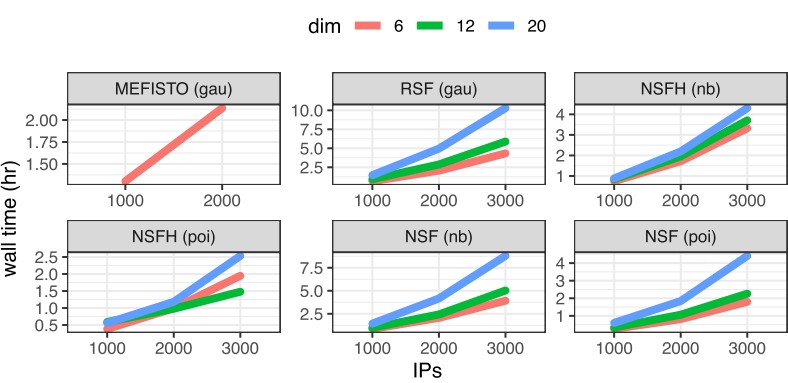

(c) Time to converge

**Extended Data Fig. 8 | Benchmarking number of inducing points (IPs) in spatial factor models on Slide-seqV2 mouse hippocampus gene expression data.** RSF: real-valued spatial factorization, NSF: nonnegative spatial factorization, NSFH: NSF hybrid model, dim: number of latent dimensions or components, gau: Gaussian, poi: Poisson, nb: negative binomial. (a) Goodness of fit increases (training deviance decreases) for increasing number of IPs in spatial models RSF and NSF with larger numbers of components. (b) No clear effect of number of IPs on predictive accuracy (validation deviance). (c) Higher numbers of IPs are more computationally expensive (time to convergence).

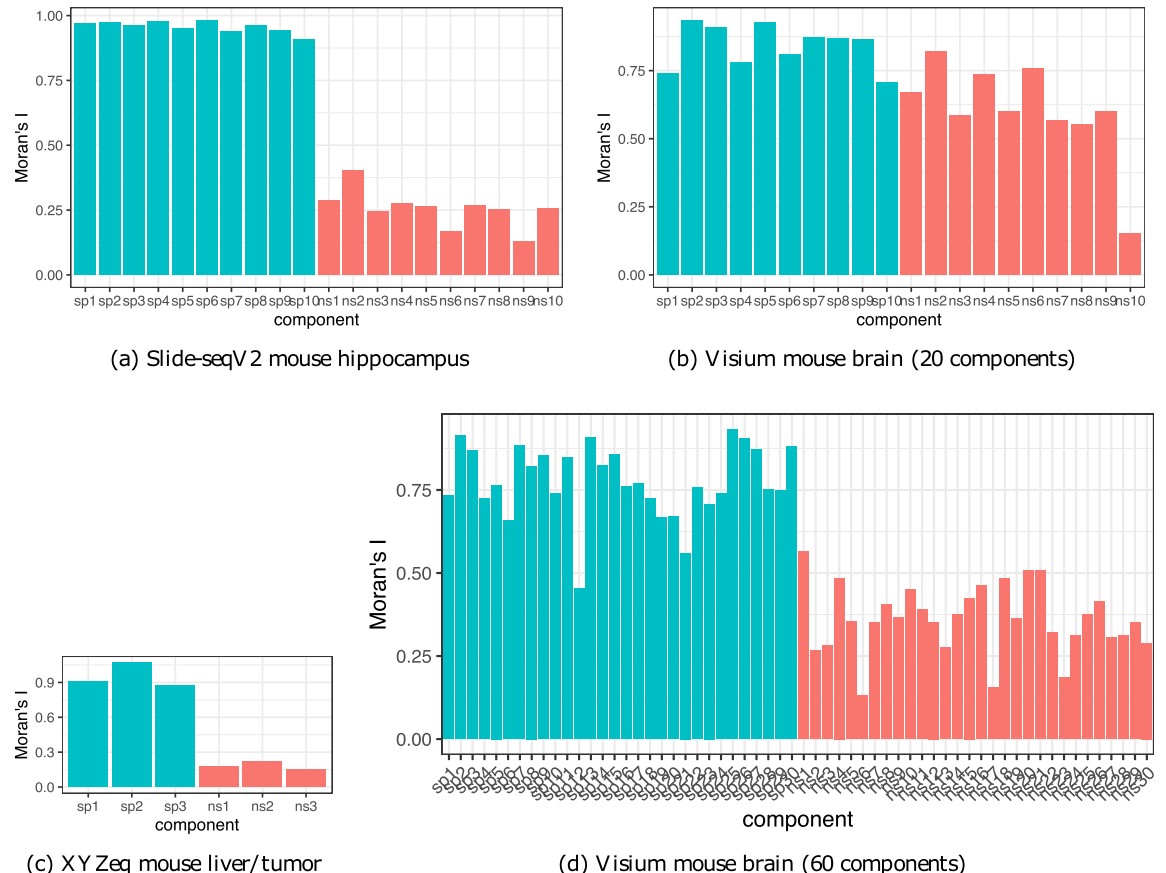

(a) Slide-seqV2 mouse hippocampus

(b) Visium mouse brain (20 components)

(c) XYZeq mouse liver/tumor

(d) Visium mouse brain (60 components)

**Extended Data Fig. 9 | Autocorrelation of spatial and nonspatial factors.** All spatial transcriptomics datasets were analyzed with the nonnegative spatial factorization hybrid model (NSFH). Blue indicates spatial factors and red indicates nonspatial factors.

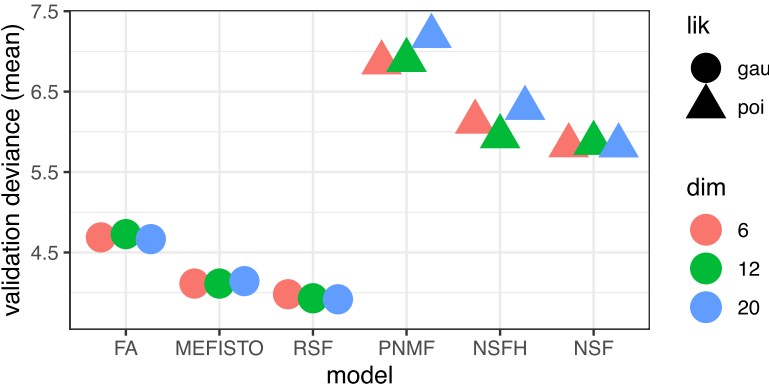

(a) Out-of-sample generalization error

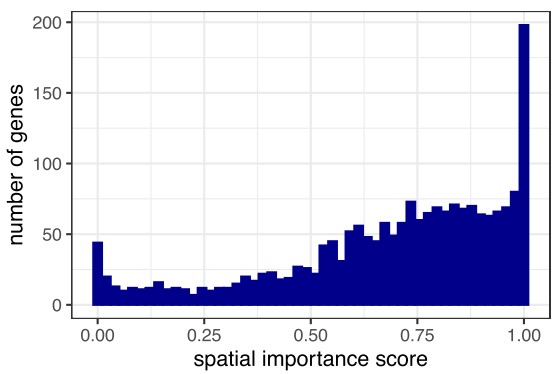

(b) Gene spatial importance

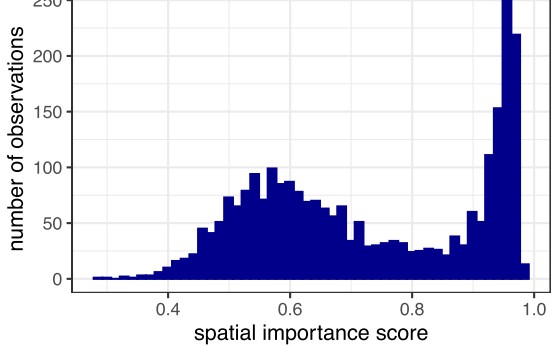

(c) Observation spatial importance

**Extended Data Fig. 10 | Benchmarking spatial and nonspatial factor models on XYZeq mouse liver gene expression data.** (a) Lower deviance indicates higher generalization accuracy. All spatial models used 288 inducing points. lik: likelihood, dim: number of latent dimensions (components), FA: factor analysis, RSF: real-valued spatial factorization, PNMF: probabilistic nonnegative matrix factorization, NSF: nonnegative spatial factorization, NSFH: NSF hybrid model. (b) Each feature (gene) was assigned a spatial importance score derived from NSFH fit with 6 components (3 spatial and 3 nonspatial). A score of 1 indicates spatial components explain all the variation. (c) as (b) but with observations instead of features.

...

# nature research

# Reporting Summary

Nature Research wishes to improve the reproducibility of the work that we publish. This form provides structure for consistency and transparency in reporting. For further information on Nature Research policies, see our Editorial Policies and the Editorial Policy Checklist.

## Statistics

For all statistical analyses, confirm that the following items are present in the figure legend, table legend, main text, or Methods section.

| n/a | Confirmed | |
|---|---|---|
| ☐ | ☒ | The exact sample size (*n*) for each experimental group/condition, given as a discrete number and unit of measurement |
| ☐ | ☒ | A statement on whether measurements were taken from distinct samples or whether the same sample was measured repeatedly |
| ☐ | ☒ | The statistical test(s) used AND whether they are one- or two-sided *Only common tests should be described solely by name; describe more complex techniques in the Methods section.* |
| ☒ | ☐ | A description of all covariates tested |
| ☐ | ☒ | A description of any assumptions or corrections, such as tests of normality and adjustment for multiple comparisons |
| ☒ | ☐ | A full description of the statistical parameters including central tendency (e.g. means) or other basic estimates (e.g. regression coefficient) AND variation (e.g. standard deviation) or associated estimates of uncertainty (e.g. confidence intervals) |
| ☒ | ☐ | For null hypothesis testing, the test statistic (e.g. *F*, *t*, *r*) with confidence intervals, effect sizes, degrees of freedom and *P* value noted *Give P values as exact values whenever suitable.* |
| ☐ | ☒ | For Bayesian analysis, information on the choice of priors and Markov chain Monte Carlo settings |
| ☒ | ☐ | For hierarchical and complex designs, identification of the appropriate level for tests and full reporting of outcomes |
| ☐ | ☒ | Estimates of effect sizes (e.g. Cohen's *d*, Pearson's *r*), indicating how they were calculated |

*Our web collection on statistics for biologists contains articles on many of the points above.*

## Software and code

Policy information about availability of computer code

| Data collection | We used Python 3.8.10 and R 4.1.0 to access publicly available datasets from previous studies. |
|---|---|
| Data analysis | We implemented all models using Python 3.8.10, tensorflow 2.5.0, and tensorflow probability 0.13.0. Other Python packages used include scanpy 1.8.0, squidpy 1.1.0, scikit-learn 0.24.2, pandas 1.2.5, numpy 1.19.5, and scipy 1.7.0. We used the MEFISTO implementation from the mofapy2 Python package, installed from the GitHub development branch at commit 8f6ffcb5b18d22b3f44ff2a06bcb92f2806afed0. Graphics were generated using either matplotlib 3.4.2 in Python or ggplot2 3.3.5 (Wickham, 2016) in R (version 4.1.0). The R packages Seurat 0.4.3 (Hao et al., 2021), SeuratData 0.2.1, and SeuratDisk 0.0.0.9019 were used for some initial data manipulations. Computationally-intensive model fitting was done on Princeton's Della cluster. Each model was assigned 12 CPU cores. We provided the following total memory per dataset: 180 Gb for Slide-seq V2, 72 Gb for Visium, and 48 Gb for XYZeq. Bioconductor (version 3.13) R packages biomaRt (version 2.48.0) and topGO (version 2.44.0) were used for gene ontology analyses. Code for reproducing the analyses of this manuscript is available from https://github.com/willtownes/nsf-paper and an installable python package is available from https://github.com/willtownes/spatial-factorization-py. |

For manuscripts utilizing custom algorithms or software that are central to the research but not yet described in published literature, software must be made available to editors and reviewers. We strongly encourage code deposition in a community repository (e.g. GitHub). See the Nature Research guidelines for submitting code & software for further information.

## Data

Policy information about availability of data

All manuscripts must include a data availability statement. This statement should provide the following information, where applicable:

- Accession codes, unique identifiers, or web links for publicly available datasets
- A list of figures that have associated raw data
- A description of any restrictions on data availability

All data used in this manuscript are from public sources. The Visium mouse brain dataset "Mouse
Brain Serial Section 1 (Sagittal-Anterior)" was downloaded from https://cf.10xgenomics.com/ samples/spatial-exp/1.1.0/V1_Mouse_Brain_Sagittal_Anterior/
V1_Mouse_Brain_Sagittal_ Anterior_filtered_feature_bc_matrix.h5 and https://cf.10xgenomics.com/samples/spatial-exp/ 1.1.0/
V1_Mouse_Brain_Sagittal_Anterior/V1_Mouse_Brain_Sagittal_Anterior_spatial.tar.
gz. The Slide-seqV2 mouse hippocampus dataset was originally produced by [9]. We obtained
it through the SeuratData R package (version 0.2.1) [28]. Raw data is available from https:// singlecell.broadinstitute.org/single_cell/study/SCP815/highly-sensitive-
spatial-transcriptomics-at study-summary. The XYZeq mouse liver data was obtained from the Gene Expression Om-
nibus [30], accession number GSE164430. We focused on sample liver slice L20C1, which was
featured in the original publication, and downloaded it as an H5AD file. The spatial coordi-
nates were provided by the original authors. Additional databases used in the study included
Biomart (http://www.biomart.org/), Panglao (https://panglaodb.se/), and scfind (https: //scfind.sanger.ac.uk/).

# Field-specific reporting

Please select the one below that is the best fit for your research. If you are not sure, read the appropriate sections before making your selection.

☒ Life sciences ☐ Behavioural & social sciences ☐ Ecological, evolutionary & environmental sciences

For a reference copy of the document with all sections, see nature.com/documents/nr-reporting-summary-flat.pdf

# Life sciences study design

All studies must disclose on these points even when the disclosure is negative.

| | |
|---|---|
| Sample size | We used only public data, so no sample size calculations were performed ahead of time. Sample sizes reflected inclusion of as many observations as possible from original studies subject to standard quality control filtering. Sample sizes for simulations were set to mimic typical values found in public datasets. |
| Data exclusions | Visium mouse brain: observations (spots) with total counts less than 100 or mitochondrial counts greater than 20% were excluded. Slide-seqV2 mouse hippocampus: observations (spots) with total counts less than 100 or mitochondrial counts greater than 20% were excluded. |
| Replication | While we did not generate any data, all code is provided openly to enable others to replicate our results. |
| Randomization | We did not generate new data for this study, so we did not randomize the data. We did use different random seeds in simulations to create a range of scenarios for assessing the relative performance of different models. |
| Blinding | Blinding was not necessary for this study, since we did not generate any new data to test a hypothesis. |

# Reporting for specific materials, systems and methods

We require information from authors about some types of materials, experimental systems and methods used in many studies. Here, indicate whether each material, system or method listed is relevant to your study. If you are not sure if a list item applies to your research, read the appropriate section before selecting a response.

## Materials & experimental systems

| n/a | Involved in the study |
|---|---|
| ☒ | ☐ Antibodies |
| ☒ | ☐ Eukaryotic cell lines |
| ☒ | ☐ Palaeontology and archaeology |
| ☒ | ☐ Animals and other organisms |
| ☒ | ☐ Human research participants |
| ☒ | ☐ Clinical data |
| ☒ | ☐ Dual use research of concern |

## Methods

| n/a | Involved in the study |
|---|---|
| ☒ | ☐ ChIP-seq |
| ☒ | ☐ Flow cytometry |
| ☒ | ☐ MRI-based neuroimaging |

