## [Peer Review File. · Nature Methods]

Peer Review Information

Manuscript Title: Nonnegative spatial factorization applied to spatial genomics

Corresponding author name(s): Barbara E. Engelhardt

Editorial Notes: None

Reviewer Comments & Decisions:

Decision Letter, initial version:
--

Dear Professor Engelhardt,

Your Article, "Nonnegative spatial factorization", has now been seen by 3 reviewers. As you will see from their comments below, although the reviewers find your work of potential interest, they have raised many important concerns. We are interested in the possibility of publishing your paper in Nature Methods, but would like to consider your response to these concerns before we reach a final decision on publication.

We therefore invite you to revise your manuscript with additional analysis and other changes to fully address all these concerns. If you have any concerns about the reviewer comments, please contact us. Considering the broad readership of our journal, please make the title (and the abstract) a bit more biologist-friendly by mentioning not just the method name but its purpose.

* include a point-by-point response to the reviewers and to any editorial suggestions

* please underline/highlight any additions to the text or areas with other significant changes to facilitate review of the revised manuscript

* address the points listed described below to conform to our open science requirements

* ensure it complies with our general format requirements as set out in our guide to authors at www.nature.com/naturemethods

* resubmit all the necessary files electronically by using the link below to access your home page

[Redacted] This URL links to your confidential home page and associated information about manuscripts you may have submitted, or that you are reviewing for us. If you wish to forward this email to co-authors, please delete the link to your homepage.

We hope to receive your revised paper within 3 months. We are very aware of the difficulties caused by the COVID-19 pandemic to the community. If you cannot send it within this time, please let us know. In this event, we will still be happy to reconsider your paper at a later date so long as nothing similar has been accepted for publication at Nature Methods or published elsewhere.

OPEN SCIENCE REQUIREMENTS

REPORTING SUMMARY AND EDITORIAL POLICY CHECKLISTS

Please note that these forms are dynamic ‘smart pdfs’ and must therefore be downloaded and completed in Adobe Reader. We will then flatten them for ease of use by the reviewers. If you would like to reference the guidance text as you complete the template, please access these flattened versions at <http://www.nature.com/authors/policies/availability.html>.

DATA AVAILABILITY

All novel DNA and RNA sequencing data, protein sequences, genetic polymorphisms, linked genotype and phenotype data, gene expression data, macromolecular structures, and proteomics data must be deposited in a publicly accessible database, and accession codes and associated hyperlinks must be provided in the “Data Availability” section.

Please include a “Data availability” subsection in the Online Methods. This section should inform readers about the availability of the data used to support the conclusions of your study, including accession codes to public repositories, references to source data that may be published alongside the paper, unique identifiers such as URLs to data repository entries, or data set DOIs, and any other statement about data availability. At a minimum, you should include the following statement: “The data that support the findings of this study are available from the corresponding author upon request”, describing which data is available upon request and mentioning any restrictions on availability. If DOIs are provided, please include these in the Reference list (authors, title, publisher (repository name), identifier, year). For more guidance on how to write this section please see: <http://www.nature.com/authors/policies/data/data-availability-statements-data-citations.pdf>

CODE AVAILABILITY

Please include a “Code Availability” subsection in the Online Methods which details how your custom code is made available. Only in rare cases (where code is not central to the main conclusions of the paper) is the statement “available upon request” allowed (and reasons should be specified).

MATERIALS AVAILABILITY

ORCID

Nature Methods is committed to improving transparency in authorship. As part of our efforts in this direction, we are now requesting that all authors identified as ‘corresponding author’ on published papers create and link their Open Researcher and Contributor Identifier (ORCID) with their account on the Manuscript Tracking System (MTS), prior to acceptance. This applies to primary research papers only. ORCID helps the scientific community achieve unambiguous attribution of all scholarly contributions. You can create and link your ORCID from the home page of the MTS by clicking on ‘Modify my Springer Nature account’. For more information please visit www.springernature.com/orcid.

Sincerely,

Lin Tang, PhD
Senior Editor
Nature Methods

Reviewers' Comments:

Reviewer #1:

Remarks to the Author:

Here Townes & Engelhardt develop count valued factor analysis models where (some) factors have Gaussian process priors based on spatial coordinates of the observed data. These models can be used to identify spatial domains in spatial transcriptomics data with sets of co-expressing genes. These factor models further enforce non-negative constraints on observation weights and analyte weights, which the authors argue provide more interpretable factorizations.

The proposed models are compared with similar models from literature using goodness-of-fit on held out data, albeit held out data from a single dataset. This measures how well the models generalize to data within a dataset, and not necessarily to novel data. Still, it represents how well the data at hand is represented by the different models.

A number of relevant models have been identified and assessed in a thorough literature review. However, one notable omission is Berglund et al 2018, where the authors create a similar model of spatial Poisson factor analysis for the analysis of spatial transcriptomics data (<https://github.com/maaskola/spatial-transcriptome-deconvolution>) from prostate cancer samples.

It still remains an unfortunate fact in the spatial transcriptomics field that there is complete lack of 'ground truth' data. Here the proposed method is applied to a number of datasets as a demonstration. It is noted that patterns inferred by the factors resemble known anatomical patterns. But the precise amount of agreement cannot be quantified.

It would be great for readers if the authors could more clearly explain why the non-negative factors help users in terms of interpretability. In the current manuscript it is stated as an obvious fact that non-negative parts-based representation is desirable, but for the audience of this journal this subtle point needs to be motivated and explained. The current example with eyes and noses vs eigenfaces in images

is not particularly clear. This becomes particularly important since the authors acknowledge that real-valued models can store more information in a smaller set of factors, which makes it seem as if real-valued factors is a preferable choice.

Similarly, it should be clarified in the main text that the simulated spatial ggblocks patterns directly correspond to spatial factors as a ground truth. (This is clarified in the methods section, but putting that information where the simulations are introduced will provide readers with more clear intuition of the model).

Since the authors know the ground truth 'spatial importance' from the M1 + M2 simulations, they should report how estimated spatial importances correspond to known spatial importances. In general the plots in Figure 1 should come paired with quantitative comparisons with plots of inferred vs observed values as well as deviances for the different models.

In the initial description of 'nonspatial models' the authors should specify how a \tilde{Y} matrix is obtained (scaling, log-transforming, mean-scaling).

Overall the paper is very thorough, and performs a fair comparison of different models with a well motivated metric. Inherent lack of ground truth spatial transcriptomics data induces a level of skepticism in evaluation of any analysis method.

Due to lack of computational resources, the software was not tested for this review.

Reviewer #2:

Remarks to the Author:

This paper introduces two novel spatially-aware models for spatial transcriptomics (ST) analysis: nonnegative spatial factorization (NSF) and nonnegative spatial factorization hybrid (NSFH). The main premise of these models is to combine nonnegative factorizations and Gaussian processes to learn a more interpretable representations of spatial transcriptomics, compared to general Gaussian processes (without nonnegativity constraints) or to nonnegative factorization (with no spatial considerations). The NSF generative model represents each cell by a latent variable, with a Gaussian process (GP) prior. To account for the spatial organization of cells, the prior relies on a Matern covariance from spatial coordinates. Count means decompose as a nonnegative factorization of the latent variables. The authors argue that this representation enforces the sparsity and interpretability of the loadings. The NSFH model is an extension of NSF. In addition to the GP latent variable, space-agnostic factors aim at characterizing non-spatial underlying factors. For inference, the approach relies on a variational approach, using the inducing points method. The authors compare these models to both space-unaware models (factor

analysis (FA) and probabilistic NMF), and space-aware approaches (real-valued spatial factorization (RSF), and MEFISTO).

Altogether, we find this to be an important and timely contribution to the growing field of ST. However, there are several issues with the most critical ones related to the merit of the suggested approach over previous ones and to its evaluation. These and additional points are listed below.

1. General comments:

1.1. Please clarify which parts of the presented methods are novel (e.g., in comparison to Schmidt and Laurberg, 2008).

1.2. Throughout the paper (with simulations and real data), we are missing estimation of statistical significance of the differences between methods. These are essential to evaluate the merit of this contribution.

1.3. It is difficult to keep track of which exact models and hyperparameters were used in the experiments. It would be helpful to include a summary table describing generative model likelihoods, kernel hyperparameters, and any other important hyperparameter.

1.4. Similarly, please provide clarity for how was the number of factors selected and provide guidance for how should it be set by prospective users. Also, please clarify how were the number of inducing points selected in the different experiments and what is the effect of too few inducing points on performance

1.5. While the authors demonstrate the merit of their choice of kernel function by comparing RSF to MEFISTO, more evidence is required to justify this choice for the nonnegative methods. Please add benchmarks for NSF/NSFH with kernels other than Matern – using both simulated and real data.

2. Simulation (Figure 1): the results of the simulation are visually appealing. However, to make a convincing case, the following should be addressed:

2.1. While the authors consider complex and overlapping spatial patterns, it seems that four patterns is somewhat lower than what is observed in real data set. Please add simulations with increasing numbers of patterns.

2.2. For better evaluation of the results, please add:

2.2.1. Quantification of accuracy of the inferred patterns (e.g., distance between inferred and simulated factors). These results should also demonstrate the variation in accuracy across many simulation runs (not only average).

2.2.2. Quantitative analysis of how well are the features assigned (via loading values) to their correct spatial pattern.

2.3. NSFH model:

2.3.1. Regarding the model's ability to distinguish between spatial and non-spatial components. Here, the authors should add purely nonspatial genes, and quantify the extent to which it is reflected by NSFH.

2.3.2. The current simulation process is presented in the main text as fully spatial (all genes are spatial). It is therefore not very clear why NSFH provides the visually cleanest results. We are assuming this may have to do with the addition of noise, but this is unclear since the M2 component is also spatial. Please clarify this point.

2.3.3. What happens if NSFH uses different numbers of factors? This will provide a more "fair" comparison to other methods; especially in the case of non-spatial ones.

3. Analysis of real data sets (Figure 2-7): The main premise of NSF/NSFH is that they could provide more interpretable factors than non-constrained loading approaches (albeit with a larger held out error) and accurately associate genes with these factors. However, we are missing quantification of these properties and a more formal comparison to other approaches. These are critical to evaluate the merit of NSF (over RSF, MEFISTO and PNMF) and NSFH (over NSF).

3.1. Please include the factors' posterior means of RSF/PNMF and FA (Figures 3, 5, and 7) to support the claim that NSFH/NSF provide better interpretability. More standard approaches should also be included in this analysis (e.g., clustering of spots and [in a separate analysis] clustering of genes [e.g., with Seurat, Scanpy, SpatialDE, or Hotspot]). Can NSFH/NSF reveal more nuanced (and relevant) patterns than those achieved by the previous/ simpler approaches?

3.2. Please add a quantification of "spatialization" of the posterior factors (e.g., using autocorrelation), to better support the discussion of spatial vs. nonspatial factors (e.g., Figure 5) and compare the methods.

3.3. Sparsity of gene loadings should be reported and compared for each method in each of the real data sets.

3.4. Several methods already exist for identification of spatially- variable genes (examples listed above). A comparison should be made between the sets of genes identified by these methods and the genes assigned with high "spatial importance" by NSFH. What is the added value of NSFH here?

3.5. Please explain what is the motivation behind Poisson deviance as a way to compare model fitting? This may not provide an accurate view of the different models' performance, especially since not all approaches use the same noise model.

3.6. Related to that - the 95%-5% split may leave too little for validation, especially for the Visium dataset. Please demonstrate how the results vary with the extend of held-out data.

3.7. For all the analysis above, whenever possible, the authors should report variance in performance as well (e.g., by holding out different sets of observations).

3.8. It is not clear how the gene enrichment analyses were conducted. Did the authors take all non-zero loading coefficients for each factor, and display the top returning gene set? How were the five genes appearing in each table selected? Please present some statistics as to how many genes are deemed "associated" with each factor.

4. Spatial vs. non spatial factors:

4.1. One of the key contribution of this paper is the hybrid (spatial, non spatial) NSFH model. One would generally expect that NSFH would provide a better fit than NSF on the real data (which should include non-spatial factors), but this is not the case. Beyond the simulation analysis (summarized above), the authors should better justify the merit of NSFH as a model to be used in practice.

4.2. In tissues with spatial mixing of cell types (which seems to be the case for the liver), one would expect cell-type specific expression programs to be captured by non spatial factors. Do we see evidence for this in the XYSeq data? For instance, if we create a density map for every cell type (which is possible, since we have single cell resolution; can also be done in Figure 7 using deconvolution), do those correlate with any of the factors

Additional comments

- In the description of probabilistic nonnegative matrix factorization, we are missing a requirement for w to be non-negative.
- It is not clear how many inducing points were used in the Slideseq2 experiment (3,000 vs 2,000 inducing points). If this number changes, why is it the case?
- Figure 1: Please clarify how the figure panels were produced for each algorithm.
- MEFISTO should be included in all of the experiments. If this is unfeasible due to OOM errors, please state so clearly.
- Please clarify how many training steps the algorithms were trained in. We do not know how learning rates were selected for the Adam optimizer for the different algorithms.
- Page 19: While it is hinted from the context that L1 corresponds to the expected log-likelihood, the term is not defined until several lines later, which creates some confusion.
- Page 20: point type in the fourth equation. Same with the one before the last.

Reviewer #3:

Remarks to the Author:

The authors present nonnegative spatial factorization (NSF) a spatially-aware nonnegative matrix factorization (NMF) encoding a Gaussian process prior over the spatial locations and with a Poisson or negative binomial likelihood for count data. They then combine this spatially-aware dimension reduction with nonspatial factors in a NSF hybrid model (NSFH) to partition variability into the spatial and nonspatial sources. Applying both methods to both simulated and real data, the authors demonstrate favorable performance compared to factor analysis, MEFISTO, probabilistic NMF. They identify appropriate GP kernels and develop inference methods for the kernel parameters and latent variables to

enable computationally tractable fitting of large field-of-view ST data. Additionally, they develop a score to compute the equivalent of percent variance explained for NMF, NSF, and NSFH components.

The code to reproduce the analysis is available. While the annotation of the code is minimal it is sufficient to reproduce all of the analyses as described in the manuscript with acceptable deviations arising from system difference and the inherent stochastic nature of some of the decomposition procedures. The manuscript is well written and the methods of sufficient novelty and value to the community that they warrant publication provided the following minor concerns are addressed.

Minor:

1. The axis labels in Figure 2, 4, and 6 are illegible.
2. The hybrid model appears to have a smoothing effect in the simulation data which is interesting, but not discussed. The advantage of the hybrid model in real data applications is obvious with regard to not spatial biological sources of variation; however, the addition of a denoising effect would make it an even stronger tool.
3. The stability of the factorization to the number of components and the effect of ratio of spatial to non-spatial components is not discussed and deserves at least a mention in the main document if not more detailed description in the methods.

Author Rebuttal to Initial comments

Response to reviewers

Thank you for the reviews for our manuscript. We have addressed each point raised by the reviewers; reviewers' comments are in bold text and our responses are in plain text.

Reviewer 1

A number of relevant models have been identified and assessed in a thorough literature review. However, one notable omission is Berglund et al 2018, where the authors create a similar model of spatial Poisson factor analysis for the analysis of spatial transcriptomics data (<https://github.com/maaskola/spatial-transcriptome-deconvolution>) from prostate cancer samples.

We now include this citation in the introduction.

It would be great for readers if the authors could more clearly explain why the non-negative factors help users in terms of interpretability. In the current manuscript it is stated as an obvious fact that non-negative parts-based representation is desirable, but for the audience of this journal this subtle point needs to be motivated and explained. The current example with eyes and noses vs eigenfaces in images is not particularly clear. This becomes particularly important since the authors acknowledge that real-valued models can store more information in a smaller set of factors, which makes it seem as if real-valued factors is a preferable choice.

We have now added additional clarification to the introduction.

Similarly, it should be clarified in the main text that the simulated spatial ggblocks patterns directly correspond to spatial factors as a ground truth. (This is clarified in the methods section, but putting that information where the simulations are introduced will provide readers with more clear intuition of the model).

We have now added a sentence in the results section to address this point.

Since the authors know the ground truth ‘spatial importance’ from the M1 + M2 simulations, they should report how estimated spatial importances correspond to known spatial importances. In general the plots in Figure 1 should come paired with quantitative comparisons with plots of inferred vs observed values as well as deviances for the different models.

We have now added quantitative benchmarking of different models in multiple replicate simulations, both to assess predictive accuracy with deviance as well as correlations between learned and true factors and loadings, and spatial importance scores. The results are summarized in two supplemental figures.

In the initial description of ‘nonspatial models’ the authors should specify how a \tilde{Y} matrix is obtained (scaling, log-transforming, mean-scaling).

We have now modified the text to include “such as a mean-centered log of counts per million”. The specific normalization and transformation procedures we used in our data analyses are described in more detail in the methods section.

Reviewer #2:

1. General comments:

1.1. Please clarify which parts of the presented methods are novel (e.g., in comparison to Schmidt and Laurberg, 2008).

Thanks for this reference, we were previously unaware of it. We have added the following to the introduction: “An important prior work by Schmidt and Laurberg, 2008 proposed

GPP-NMF, which is NMF with GP priors. Our approach differs from GPP-NMF in that we use variational inference rather than maximum a posteriori point estimation, our model can flexibly handle large numbers of irregularly spaced or missing spatial observations, and we automatically learn all hyperparameters during model fitting rather than manually tuning them.”

1.2. Throughout the paper (with simulations and real data), we are missing estimation of statistical significance of the differences between methods. These are essential to evaluate the merit of this contribution.

We now report test statistics and p-values for significance tests (linear regressions and t-tests) throughout the results section.

1.3. It is difficult to keep track of which exact models and hyperparameters were used in the experiments. It would be helpful to include a summary table describing generative model likelihoods, kernel hyperparameters, and any other important hyperparameter.

We have now added additional clarifying text throughout the results to address this. Specifically we note that the Poisson likelihood is always used for nonnegative models and the Gaussian likelihood for real-valued models unless otherwise mentioned. Note that all kernel hyperparameters are learned by gradient-based optimization (Adam) so we do not list their values explicitly.

1.4. Similarly, please provide clarity for how was the number of factors selected and provide guidance for how should it be set by prospective users. Also, please clarify how were the number of inducing points selected in the different experiments and what is the effect of too few inducing points on performance

We have now added a paragraph in the discussion to address the number of factors. In the simulation studies and for XYZeq and Visium, we always set the number of IPs equal to the number of observations. We have clarified this in figure captions and in the results. For Slide-seqV2, the number of observations was too large so we varied the number of IPs to examine the effect on performance in the benchmarking. We have added a new paragraph to this section discussing the results along with a new supplemental figure.

1.5. While the authors demonstrate the merit of their choice of kernel function by comparing RSF to MEFISTO, more evidence is required to justify this choice for the nonnegative methods. Please add benchmarks for NSF/NSFH with kernels other than Matern – using both simulated and real data.

We re-ran the benchmarking on the Visium brain dataset with RSF, NSF, and NSFH using exponentiated quadratic (aka squared exponential) kernels, which is the same as used by MEFISTO. We were surprised to find that our models still outperformed MEFISTO with M=1000 inducing points:

The small differences in performance between the same models fitted with different kernels suggest our models are more robust to kernel choice than we originally thought. This indicates that the difference between MEFISTO and RSF must be due to some other factors. Some possibilities include:

1. Coordinate ascent optimization in MEFISTO versus momentum-based gradient descent with Adam for RSF.
2. Lack of a sparsity promoting prior in RSF
3. RSF includes a linear model as a mean function, with learnable parameters for each latent GP whereas MEFISTO has zero mean
4. Possibly other implementation details related to eg stopping conditions.

Another difference is that MEFISTO restricts the kernel to be isotropic (ie to have the same hyperparameters for each of the spatial directions). While all of our spatial factor model implementations support more flexible anisotropic kernels, in practice we used isotropic kernels in all our experiments to facilitate comparisons.

We have now included the above as a supplemental figure and added a paragraph in the visium results section to discuss.

2. Simulation (Figure 1):

The results of the simulation are visually appealing. However, to make a convincing case, the following should be addressed:

2.1. While the authors consider complex and overlapping spatial patterns, it seems that four patters is somewhat lower than what is observed in real data set. Please add simulations with increasing numbers of patterns.

We have now included a combined simulation that includes both the quilt and blocks patterns (8 total spatial patterns). Benchmarking of the models on replicates of each simulation is shown

below (metric is predictive accuracy on held-out data). This is now in the manuscript as a supplemental figure.

2.2. For better evaluation of the results, please add:

2.2.1. Quantification of accuracy of the inferred patterns (e.g., distance between inferred and simulated factors). These results should also demonstrate the variation in accuracy across many simulation runs (not only average).

We now include this assessment as a supplemental figure.

2.2.2. Quantitative analysis of how well are the features assigned (via loading values) to their correct spatial pattern.

We now include this assessment as a supplemental figure.

2.3. NSFH model:

2.3.1. Regarding the model's ability to distinguish between spatial and non-spatial components. Here, the authors should add purely nonspatial genes, and quantify the extent to which it is reflected by NSFH.

We now include this assessment as a supplemental figure.

2.3.2. The current simulation process is presented in the main text as fully spatial (all genes are spatial). It is therefore not very clear why NSFH provides the visually cleanest results. We are assuming this may have to do with the addition of noise, but this is unclear since the M2 component is also spatial. Please clarify this point.

We decided to rewrite this section with a different, simpler and easier to explain simulation paradigm using only the 4 spatial factors (M1) and getting rid of the (M2) nonspatial components. Originally the simulation included both spatial (4) and nonspatial (3) factors but we fit all models except NSFH with L=4 factors. The nonspatial factors served as essentially low-rank noise to make model fitting more challenging for each method. NSFH was able to “denoise” by including the additional 3 nonspatial factors. However, we now realize this was

not a fair comparison as it's difficult to tell whether the denoising is due to the spatial/nonspatial nature of the factors or due to simply a larger number of total factors. We have now removed NSFH from the illustrative examples as it is not directly comparable to the other models in this case. The parts-based representation is still apparent in the other nonnegative models PNMf and NSF. We now provide a separate assessment of NSFH described below.

2.3.3. What happens if NSFH uses different numbers of factors? This will provide a more “fair” comparison to other methods; especially in the case of non-spatial ones.

We now include a quantitative benchmarking of NSFH versus PNMf and NSF with the same total number of factors. PNMf is a special case of NSFH where the number of spatial factors is zero, and NSF is a special case of NSFH where the number of spatial factors is equal to the total number of factors. The following is now a supplemental figure and discussed in the results section, it shows that NSFH has better predictive accuracy than NSF and PNMf with the same total number of factors on the simulated data, and that it is much more accurate in estimating spatial importance scores.

3. Analysis of real data sets (Figure 2-7):

The main premise of NSF/NSFH is that they could provide more interpretable factors than non-constrained loading approaches (albeit with a larger held out error) and accurately associate genes with these factors. However, we are missing quantification of these properties and a more formal comparison to other approaches. These are critical to evaluate the merit of NSF (over RSF, MEFISTO and PNMF) and NSFH (over NSF).

3.1. Please include the factors' posterior means of RSF/PNMF and FA (Figures 3, 5, and 7) to support the claim that NSFH/NSF provide better interpretability. More standard approaches should also be included in this analysis (e.g., clustering of spots and [in a separate analysis] clustering of genes [e.g., with Seurat, Scanpy, SpatialDE, or Hotspot]). Can NSFH/NSF reveal more nuanced (and relevant) patterns than those achieved by the previous/ simpler approaches?

Clustering of observations. We have now included heatmaps of FA, RSF, PNMF, and scanpy clustering of observations for each of the three datasets as supplemental figures. We have also included scanpy clustering in the simulations heatmaps.

Clustering of features. We applied hotspot to each dataset and matched the top genes on each cluster to GO biological process terms. We include the results as supplemental tables and refer to them in the results.

3.2. Please add a quantification of “spatialization” of the posterior factors (e.g., using autocorrelation), to better support the discussion of spatial vs. nonspatial factors (e.g., Figure 5) and compare the methods.

We now include Moran's I statistics for each of the spatial and nonspatial components as supplemental figures and refer to them in the results.

3.3. Sparsity of gene loadings should be reported and compared for each method in each of the real data sets.

We now include this as a supplemental figure.

3.4. Several methods already exist for identification of spatially- variable genes (examples listed above). A comparison should be made between the sets of genes identified by these methods and the genes assigned with high “spatial importance” by NSFH. What is the added value of NSFH here?

We now compare Hotspot's list of spatially variable genes to the results of NSFH. We define a Hotspot variable gene as having an FDR-adjusted p-value less than 0.05. In all three datasets, Hotspot labeled all 2000 variable genes as spatially variable. We define a NSFH variable gene as having a spatial importance score of greater than 0.5. NSFH labeled 89, 412, and 19 genes as nonspatial in each of the three datasets, respectively. We also computed Spearman correlations between the Hotspot Z-scores and NSFH spatial importance scores and report them in the results.

3.5. Please explain what is the motivation behind Poisson deviance as a way to compare model fitting? This may not provide an accurate view of the different models' performance, especially since not all approaches use the same noise model.

We have now included comparisons of models on the three real datasets using root mean squared error as a complement to the use of Poisson deviance.

3.6. Related to that - the 95%-5% split may leave too little for validation, especially for the Visium dataset. Please demonstrate how the results vary with the extend of held-out data.

We repeated the analyses used to produce Figure 6a with a 80%/20% split. While this led to uniformly higher generalization error estimates (ie worse predictive performance) across all models, the relative performance between models was unchanged.

Original figure 6a with 95/5 split

Version with 80/20 split (now included as a supplemental figure)

3.7. For all the analysis above, whenever possible, the authors should report variance in performance as well (e.g., by holding out different sets of observations).

We now include multiple replicates of each simulation, where each replicate has a different set of held-out observations. We have now included regression analyses to quantify the statistical significance of differences in models, which takes into account variability across model runs. We now also include a supplemental figure comparing the validation deviance with a larger set of observations held out versus a smaller set (ie the previous comment).

3.8. It is not clear how the gene enrichment analyses were conducted. Did the authors take all non-zero loading coefficients for each factor, and display the top returning gene set? How were the five genes appearing in each table selected? Please present some statistics as to how many genes are deemed “associated” with each factor.

We explain this in the methods section “Cell types and GO terms”. Each component is represented by a column in the loadings matrix. For each component, we can rank genes according to their nonnegative loading weight. The five genes in the tables are those with the highest weights and we used these to search for cell types. For GO analysis we take the top 100 genes for each component and use all genes as the “background”.

4. Spatial vs. non spatial factors:

4.1. One of the key contribution of this paper is the hybrid (spatial, non spatial) NSFH model. One would generally expect that NSFH would provide a better fit than NSF on the real data (which should include non-spatial factors), but this is not the case. Beyond the simulation analysis (summarized above), the authors should better justify the merit of NSFH as a model to be used in practice.

We now include a supplemental figure showing the goodness-of-fit of NSFH versus NSF to the training data for each dataset using Poisson deviance as a metric. For Slide-seqV2 and XYZeq, NSFH has better fit to training data than NSF. For Visium NSF outperforms with a negative binomial likelihood but they are similar with the Poisson likelihood. The main advantage of NSFH over NSF however is to be able to interpret and compare nonspatial versus spatial factors within a single modeling framework.

4.2. In tissues with spatial mixing of cell types (which seems to be the case for the liver), one would expect cell-type specific expression programs to be captured by non spatial factors. Do we see evidence for this in the XYZeq data? For instance, if we create a density map for every cell type (which is possible, since we have single cell resolution; can also be done in Figure 7 using deconvolution), do those correlate with any of the factors

We now include a supplemental figure with heatmaps of all the cell types annotated by the original authors in the XYZeq dataset.

Additional comments

- **In the description of probabilistic nonnegative matrix factorization, we are missing a requirement for w to be non-negative.**

We now include this in the text.

- **It is not clear how many inducing points were used in the Slideseq2 experiment (3,000 vs 2,000 inducing points). If this number changes, why is it the case?**

We were not able to fit MEFISTO with 3,000 inducing points so we fit all other models with 2,000 to provide a fair comparison in the benchmarking. For the more detailed exploratory analysis of the dataset we used as many IPs as was computationally feasible (3,000).

- **Figure 1: Please clarify how the figure panels were produced for each algorithm.**

We now provide a detailed explanation of how each model was fit in the methods section.

- **MEFISTO should be included in all of the experiments. If this is unfeasible due to OOM errors, please state so clearly.**

We now include a sentence in the figure legend for the slide-seq2 experiment to clarify that we could not run MEFISTO with more than 6 components due to OOM error.

- **Please clarify how many training steps the algorithms were trained in. We do not know how learning rates were selected for the Adam optimizer for the different algorithms.**

We now explain how the learning rates were set and how we detected convergence to stop optimization in the methods section.

- **Page 19: While it is hinted from the context that L1 corresponds to the expected log-likelihood, the term is not defined until several lines later, which creates some confusion.**

We now define this symbol immediately below the equation where it first occurs.

- **Page 20: point type in the fourth equation. Same with the one before the last.**

This should now be fixed.

Reviewer #3:

Minor:

1. **The axis labels in Figure 2, 4, and 6 are illegible.**

We have now reformatted these figures to improve legibility.

2. **The hybrid model appears to have a smoothing effect in the simulation data which is interesting, but not discussed. The advantage of the hybrid model in real data applications is obvious with regard to not spatial biological sources of variation; however, the addition of a denoising effect would make it an even stronger tool.**

We have now removed NSFH from this simulation because we realized it had an unfair advantage over the other models- NSFH had 4 spatial factors and 3 nonspatial factors, whereas the others only had 4 total factors. We now include denoising as a possible future application of NSFH in the discussion section.

3. **The stability of the factorization to the number of components and the effect of ratio of spatial to non-spatial components is not discussed and deserves at least a mention in the main document if not more detailed description in the methods.**

We now include supplemental figures comparing the various models with different numbers of components. We also include a new simulation where we compare NSFH to NSF (a special case where all components are spatial) and PNMF (a special case where all components are nonspatial) to assess the importance of choosing the right number of spatial vs nonspatial components.

Decision Letter, first revision:

Dear Dr. Engelhardt,

Thank you for submitting your revised manuscript "Nonnegative spatial factorization" (NMETH-A47325A). It has now been seen by the original referees and their comments are below. The reviewers find that the paper has improved in revision, and therefore we'll be happy in principle to publish it in Nature Methods, pending minor revisions to satisfy the referees' final requests and to comply with our editorial and formatting guidelines.

To make this study more accessible to our readers with biological expertise, please revise the title to hint at the biological applications of nonnegative spatial factorization.

TRANSPARENT PEER REVIEW

Nature Methods offers a transparent peer review option for new original research manuscripts submitted from 17th February 2021. We encourage increased transparency in peer review by publishing the reviewer comments, author rebuttal letters and editorial decision letters if the authors agree. Such peer review material is made available as a supplementary peer review file. Please state in the cover letter 'I wish to participate in transparent peer review' if you want to opt in, or 'I do not wish to participate in transparent peer review' if you don't. Failure to state your preference will result in delays in accepting your manuscript for publication.

Thank you again for your interest in Nature Methods Please do not hesitate to contact me if you have any questions.

Sincerely,

Lin Tang, PhD
Senior Editor
Nature Methods

ORCID

Reviewer #1 (Remarks to the Author):

In this revision Townes & Engelhardt has added a great number of quantitative results comparing model fits to fitted and observed data. These inclusions have addressed the issues raised in the original submission regarding the proposed spatial non-negative factorisation model proposed by the authors.

Reviewer #2 (Remarks to the Author):

Our comments were largely addressed and we believe that the manuscript in its current form will provide an important contribution to the field.

Remaining minor points:

1. Regarding the choice of Kernel function. The authors now find that the Matern and exponentiated quadratic kernels (used in a previous study) perform comparably when plugged into their method. This needs to be better clarified and the reasons for the different performance between methods (as they are provided in the response for reviewers) should be clearly outlined (ideally, in the discussion section). Related to that, it is stated that “Matern kernel to have better numerical stability than the EQ in our experiments.” Please be sure that this result is demonstrated in the manuscript.
2. NSFH predictions of spatially- related genes are now compared to the Hotspot algorithm. The authors demonstrate that the way the genes are stratified into groups is more sensible in NSFH using enrichment analysis. However, there is no indication whether the selection of genes per-se works better

with NSFH (it is unclear what the correlation between NSFH and Hotspot scores [of lack thereof] means). We see that NSFH calls less genes than Hotspot. Can the authors demonstrate that the genes called by the latter and not the former are indeed false positives? (e.g., through specific examples of genes or gene clusters)

Reviewer #3 (Remarks to the Author):

The authors have addressed my concerns. I am excited to see this paper published as I think it will be useful to the field.

Author Rebuttal, first revision:

Response to reviewers

Thank you for the reviews of our manuscript. We have addressed each point raised by the editors and reviewers. Editorial and reviewer comments are in bold text and our responses are in plain text.

To make this study more accessible to our readers with biological expertise, please revise the title to hint at the biological applications of nonnegative spatial factorization.

We have now revised the title to “Nonnegative spatial factorization applied to genomics”.

1. Regarding the choice of Kernel function. The authors now find that the Matern and exponentiated quadratic kernels (used in a previous study) perform comparably when plugged into their method. This needs to be better clarified and the reasons for the different performance between methods (as they are provided in the response for reviewers) should be clearly outlined (ideally, in the discussion section).

We now provide this information as a new paragraph in the Discussion section.

Related to that, it is stated that “Matern kernel to have better numerical stability than the EQ in our experiments.” Please be sure that this result is demonstrated in the manuscript.

We now include a supplemental table summarizing convergence of different models, likelihoods, number of inducing points, and kernels on the Visium brain dataset.

2. NSFH predictions of spatially- related genes are now compared to the Hotspot algorithm. The authors demonstrate that the way the genes are stratified into groups is more sensible in NSFH using enrichment analysis. However, there is no indication whether the selection of genes per-se works better with NSFH (it is unclear what the correlation between NSFH and Hotspot scores [of lack thereof] means). We see that NSFH calls less genes than Hotspot. Can the authors demonstrate that the genes called by the latter and not the former are indeed false positives? (e.g., through specific examples of genes or gene clusters)

We now include in a supplemental figure: heatmaps of several genes that were assigned low spatial importance scores by NSFH, but were called as spatially variable by Hotspot.

Final Decision Letter:

Dear Professor Engelhardt,

I am pleased to inform you that your Article, "Nonnegative spatial factorization applied to spatial genomics", has now been accepted for publication in Nature Methods. Your paper is tentatively scheduled for publication in our January print issue, and will be published online prior to that. The received and accepted dates will be 11th Oct 2021 and 17th Oct 2022. This note is intended to let you know what to expect from us over the next month or so, and to let you know where to address any further questions.

Once your paper is typeset, you will receive an email with a link to choose the appropriate publishing options for your paper and our Author Services team will be in touch regarding any additional information that may be required.

Please note that *Nature Methods* is a Transformative Journal (TJ). Authors may publish their research with us through the traditional subscription access route or make their paper immediately open access through payment of an article-processing charge (APC). Authors will not be required to make a final decision about access to their article until it has been accepted. [Find out more about Transformative Journals](https://www.springernature.com/gp/open-research/transformative-journals)

Your paper will now be copyedited to ensure that it conforms to Nature Methods style. Once proofs are generated, they will be sent to you electronically and you will be asked to send a corrected version within 24 hours. It is extremely important that you let us know now whether you will be difficult to contact over the next month. If this is the case, we ask that you send us the contact information (email, phone and fax) of someone who will be able to check the proofs and deal with any last-minute problems.

If, when you receive your proof, you cannot meet the deadline, please inform us at rjsproduction@springernature.com immediately.

Once your manuscript is typeset and you have completed the appropriate grant of rights, you will receive a link to your electronic proof via email with a request to make any corrections within 48 hours. If, when you receive your proof, you cannot meet this deadline, please inform us at rjsproduction@springernature.com immediately.

Once your paper has been scheduled for online publication, the Nature press office will be in touch to confirm the details.

Content is published online weekly on Mondays and Thursdays, and the embargo is set at 16:00 London time (GMT)/11:00 am US Eastern time (EST) on the day of publication. If you need to know the exact publication date or when the news embargo will be lifted, please contact our press office after you have submitted your proof corrections. Now is the time to inform your Public Relations or Press Office about your paper, as they might be interested in promoting its publication. This will allow them time to prepare an accurate and satisfactory press release. Include your manuscript tracking number NMETH-A47325B and the name of the journal, which they will need when they contact our office.

About one week before your paper is published online, we shall be distributing a press release to news organizations worldwide, which may include details of your work. We are happy for your institution or funding agency to prepare its own press release, but it must mention the embargo date and Nature Methods. Our Press Office will contact you closer to the time of publication, but if you or your Press Office have any inquiries in the meantime, please contact press@nature.com.

Nature Portfolio journals [encourage authors to share their step-by-step experimental protocols](https://www.nature.com/nature-research/editorial-policies/reporting-standards#protocols) on a protocol sharing platform of their choice. Nature Portfolio's Protocol Exchange is a free-to-use and open resource for protocols; protocols deposited in Protocol Exchange are citable and can be linked from the published article. More details can found at www.nature.com/protocolexchange/about.

Please feel free to contact me if you have questions about any of these points. Thank you very much again for publishing your paper at Nature Methods!

Best regards,

Lin Tang, PhD
Senior Editor
Nature Methods